# CaBP1 and 2 enable sustained Ca$_V$1.3 calcium currents and synaptic transmission in inner hair cells

David Oestreicher[1,2], Shashank Chepurwar[3,4], Kathrin Kusch[5,6], Vladan Rankovic[6,7], Sangyong Jung[6†], Nicola Strenzke[3,4], Tina Pangrsic[1,2,4,8]*

[1]Experimental Otology Group, InnerEarLab, Department of Otolaryngology, University Medical Center Göttingen, Göttingen, Germany; [2]Auditory Neuroscience Group, Max Planck Institute for Multidisciplinary Sciences, Göttingen, Germany; [3]Auditory Systems Physiology Group, Institute for Auditory Neuroscience, InnerEarLab, University Medical Center Göttingen, Göttingen, Germany; [4]Collaborative Research Center 889, University of Göttingen, Göttingen, Germany; [5]Functional Auditory Genomics, Institute for Auditory Neuroscience, University Medical Center Göttingen, Göttingen, Germany; [6]Institute for Auditory Neuroscience and InnerEarLab, University Medical Center Göttingen, Göttingen, Germany; [7]Restorative Cochlear Genomics Group, Auditory Neuroscience and Optogenetics Laboratory, German Primate Cente, Göttingen, Germany; [8]Multiscale Bioimaging Cluster of Excellence (MBExC), University of Göttingen, Göttingen, Germany

*For correspondence: tpangrs@gwdg.de

Present address: †Department of Medical Science, College of Medicine, CHA University, Seong-nam-si, Gyeonggi-do, South Korea

Competing interest: The authors declare that no competing interests exist.

## eLife assessment

This **fundamental** work substantially advances our understanding of the role of calcium-binding proteins 1 and 2 (CaBP1 and CaBP2) for generating sustained calcium currents in mouse inner hair cells and their capacity for indefatigable exocytosis. The evidence supporting the conclusions is **compelling**, with rigorous in vitro and in vivo physiological experiments and state-of-the-art microscopy. The work will be of broad interest to synaptic physiologists, cellular biochemists, and hearing researchers.

**Abstract** To encode continuous sound stimuli, the inner hair cell (IHC) ribbon synapses utilize calcium-binding proteins (CaBPs), which reduce the inactivation of their Ca$_V$1.3 calcium channels. Mutations in the *CABP2* gene underlie non-syndromic autosomal recessive hearing loss DFNB93. Besides CaBP2, the structurally related CaBP1 is highly abundant in the IHCs. Here, we investigated how the two CaBPs cooperatively regulate IHC synaptic function. In *Cabp1/2* double-knockout mice, we find strongly enhanced Ca$_V$1.3 inactivation, slowed recovery from inactivation and impaired sustained exocytosis. Already mild IHC activation further reduces the availability of channels to trigger synaptic transmission and may effectively silence synapses. Spontaneous and sound-evoked responses of spiral ganglion neurons in vivo are strikingly reduced and strongly depend on stimulation rates. Transgenic expression of CaBP2 leads to substantial recovery of IHC synaptic function and hearing sensitivity. We conclude that CaBP1 and 2 act together to suppress voltage- and calcium-dependent inactivation of IHC Ca$_V$1.3 channels in order to support sufficient rate of exocytosis and enable fast, temporally precise and indefatigable sound encoding.

## Introduction

Calcium influx through voltage-gated calcium channels ($Ca_V$s) is a pivotal initiator of neurotransmission. These channels typically undergo considerable $Ca^{2+}$- and voltage-dependent inactivation (CDI and VDI), which limits the amount of incoming $Ca^{2+}$ ions and thus possible excitotoxicity. In sensory receptor cells such as sensory inner hair cells (IHCs) and photoreceptors, however, the extent of inactivation is significantly reduced to allow encoding of graded and sustained membrane potential fluctuations in response to ongoing sound or visual (t.i. 'darkness') stimuli (reviewed in *Pangrsic et al., 2018*). Furthermore, slowed inactivation of L-type $Ca_V$s was also reported at ribbon synapses of retinal bipolar cells (*von Gersdorff and Matthews, 1996*), which transmit visual information via continuous exocytosis of synaptic vesicles onto retinal ganglion cells. The attenuation of channel inactivation may be largely attributed to the action of calcium-binding proteins, CaBPs (*Haeseleer et al., 2000*), which primarily abate the calmodulin (CaM)-mediated CDI of $Ca_V$ channels (*Haeseleer et al., 2000*; *Findeisen and Minor, 2010*; *Haeseleer et al., 2004*; *Lee et al., 2002*; *Yang et al., 2006*; *Yang et al., 2014*). At the IHC ribbon synapses, the $Ca^{2+}$ current is carried through the L-type $Ca_V$1.3 channels (*Baig et al., 2011*; *Platzer et al., 2000*). Their biophysical properties may be regulated by several CaBP family members (*Cui et al., 2007*; *Yang et al., 2018*), but most prominently by CaBP2 (*Picher, 2017*). Mutations in the *CaBP2* gene cause moderate to severe autosomal recessive hearing impairment DFNB93 (see e.g. *Picher, 2017*; *Koohiyan, 2019*; *Schrauwen et al., 2012*; *Sheyanth et al., 2021*; *Tabatabaiefar et al., 2011*). Interestingly, the loss of CaBP2 seems to significantly affect VDI, but not CDI of the IHC $Ca^{2+}$ currents (*Picher, 2017*). Furthermore, isolated disruption of CaBP1 or CaBP4 in mice leads to at most modest enhancement of IHC $Ca_V$1.3 inactivation (*Cui et al., 2007*; *Yang et al., 2018*). Moreover, the automodulatory C-terminal domain of the long $Ca_V$1.3 isoform, abundant at the IHC ribbon synapses (*Bock et al., 2011*; *Scharinger et al., 2015*), does not seem critical for CDI suppression either, as its disruption does not lead to hearing impairment (*Scharinger et al., 2015*). The question of which mechanisms are primarily responsible for suppressing IHC $Ca_V$1.3 CDI thus remains unresolved. Since CaBP2, followed by the structurally similar CaBP1, are the most abundant members of the CaBP family in the cochlea (*Yang et al., 2016*), we set out to investigate the phenotype of *Cabp1/2* double-knockout (DKO) animals at the systems and cellular levels. Severe hearing impairment that we observed in these animals reveals that CaBP1 and 2 act together to attenuate the inactivation of the IHC $Ca_V$1.3 channels. Furthermore, we demonstrate that the combined absence of CaBP1 and 2 almost abolishes sustained, and, upon increased hair cell synaptic activity, the fast component of exocytosis. Such phenotype was observed neither in *Cabp1* nor in *Cabp2* single KO animals (*Yang et al., 2018*; *Picher, 2017*). To obtain a more comprehensive understanding of the roles of the two calcium-binding proteins, we attempted to rescue the auditory function of the CaBP1/2-depleted auditory system by viral delivery of a *Cabp2* coding sequence in the inner ear. These experiments revealed a substantial recovery of hearing, which may be related to the successful restoration of $Ca^{2+}$ currents and IHC exocytosis upon transgenic expression of CaBP2. Our findings indicate that the functions of CaBP1 and 2 partially overlap at the IHC ribbon synapse. Together, they provide sufficient attenuation of channel inactivation to sustain $Ca^{2+}$ currents and exocytosis required for temporally precise, indefatigable neurotransmission at this synapse.

## Results

### Pronounced VDI and CDI of $Ca_V$1.3 channels in CaBP1/2-deficient IHCs

CaBP1 and 2 show high expression levels in the organ of Corti (*Yang et al., 2016*), where they may modulate voltage-gated $Ca_V$1.3 channels at the IHC ribbon synapses (*Figure 1A*). Deletion of CaBP2 results in an early-onset, progressive hearing impairment in the KO mouse model and moderate to severe hearing impairment, DFNB93, in affected human patients (*Picher, 2017*; *Koohiyan, 2019*; *Schrauwen et al., 2012*). Loss of CaBP1 in mice, on the other hand, leaves hearing unperturbed in the first few months after birth (*Yang et al., 2018*). Since these proteins show a high degree of homology (*Haeseleer et al., 2000*), it is plausible that they possess similar features and are partially redundant, at least initially. Therefore, we were interested in understanding their cumulative impact on the function of hair cells and hearing. For this, we analyzed the phenotype of the *Cabp1/2*-DKO animals and tested the extent of functional rescue following the transgenic expression of CaBP2 (*Figure 1—figure supplement 1A*) in the ears of the DKO mice.

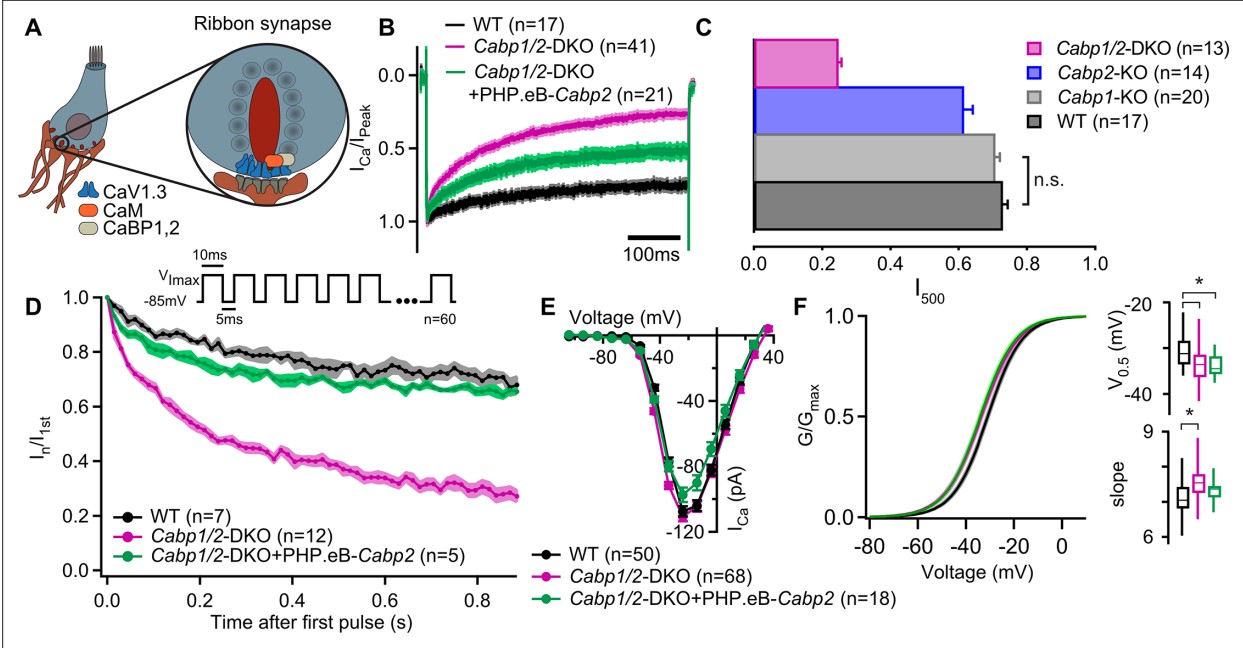

**Figure 1.** Strong inactivation of inner hair cell (IHC) $Ca_V1.3$ channels upon deletion of *Cabp1* and *Cabp2*. (**A**) Schematic of the IHC and one of its several ribbon synapses. CaBPs and CaM modulate voltage-gated calcium channels at the IHC ribbon synapses to shape the presynaptic $Ca^{2+}$ signal. (**B**) Peak normalized $Ca^{2+}$ currents after a 500-ms depolarization step to the maximum current potential in the apical IHCs of WT and *Cabp1/2*-DKO animals. Note a pronounced inactivation of the $Ca^{2+}$ current in the absence of CaBP1 and 2 and partial recovery upon *Cabp2* gene replacement. (**C**) The fraction of the remaining $Ca^{2+}$ current after a 500-ms depolarization step is depicted for different mouse models acquired at comparable conditions (room temperature, 2 mM extracellular $[Ca^{2+}]$). The histogram combines the results of this study (*Cabp1/2*-DKO) with previously published data on CaBP1 (*Yang et al., 2018*) and CaBP2 (*Picher, 2017*). The WT data combines pooled data from the current study and controls as obtained previously (*Yang et al., 2018*). Note that the recordings from *Cabp2*-KO animals were acquired at the lower apex to mid-cochlear tonotopic positions as the phenotype in the apex (low-frequency positions investigated otherwise) is mild (*Picher, 2017*). (**D**) $Ca^{2+}$ currents upon a train of 10-ms long depolarization steps normalized to the amplitude of the first pulse. (**E**) $Ca^{2+}$-current–voltage relationships show slightly reduced amplitudes in the IHCs of the *Cabp1/2*-DKO animals transduced with the PHP.eB-*Cabp2*. (**F**) Activation curves were calculated with Boltzmann fits. Note a significant difference (*asterisks*, p < 0.0001, Student's *t*-test) in the half-voltage activation ($V_{0.5}$) and slope in the CaBP1/2-deficient IHCs and the persistence of the voltage shift upon re-expression of CaBP2 (*inserts*). See main text for the p values. Data in B and D–F was acquired in 1.3 mM, data in C in 2 mM $[Ca]^{2+}$.

The online version of this article includes the following source data and figure supplement(s) for figure 1:

**Source data 1.** Traces and values obtained from patch-clamp recordings and custom-written macros that were used for analyses in *Figure 1B–D*, *Figure 2*, and *Figure 1—figure supplement 1*.

**Source data 2.** Traces and values obtained from patch-clamp recordings and custom-written macros that were used for analyses in *Figure 1E, F*.

**Figure supplement 1.** Partial rescue of $Ca_V1.3$ channel inactivation properties upon transgenic expression of CaBP2.

To investigate possible synergistic functions of the two major IHC CaBPs, we first measured $Ca^{2+}$ currents in the IHCs of 3- to 4-week-old *Cabp1/2*-DKO and WT animals in the presence of 1.3 mM extracellular $[Ca^{2+}]$. Employing 500-ms long depolarization steps to maximal $Ca^{2+}$-current potential (−25 to −15 mV) revealed pronounced inactivation of $Ca_V1.3$ channels in the DKO IHCs (*Figure 1B*). This was significantly stronger when compared to the inactivation observed in the IHCs of *Cabp1* or *Cabp2* single KOs in similar conditions (*Figure 1C*; *Yang et al., 2018*; *Picher, 2017*; *Oestreicher et al., 2021*), and even when comparing to data from 5-week-old *Cabp2* KO mice obtained at physiological temperature (*Figure 1—figure supplement 1D*), further substantiating the evidence of severity and early onset of the DKO phenotype.

The 'non-inactivating' nature of the IHC $Ca_V1.3$ channels was partially restored in the *Cabp1/2*-DKOs by transgenic expression of WT CaBP2 (Student's *t*-test: p < 0.0001; *Figure 1B, C*; see also *Figure 1—figure supplement 1*). The lack of complete recovery suggests either an inadequate expression of transgenic material or irreversible adaptive changes in response to the lack of major CaBPs that may occur prior to gene delivery. Similarly, $Ca^{2+}$ currents were greatly reduced upon application of short train stimuli and rescued efficiently by the introduction of the transgenic CaBP2 (*Figure 1D*).

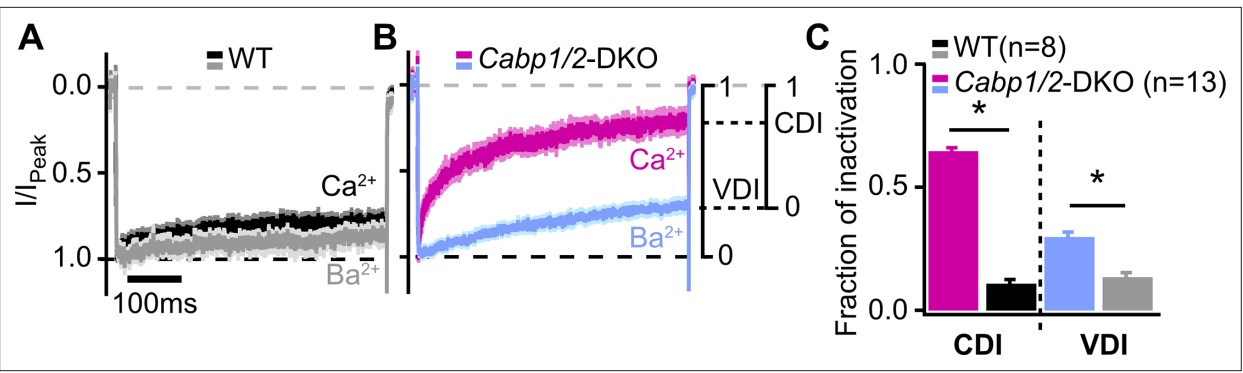

**Figure 2.** Voltage- and Ca²⁺-dependent inactivation (VDI and CDI) are enhanced after deletion of *Cabp1* and *Cabp2*. (**A, B**) Peak normalized Ca²⁺ and Ba²⁺ currents in WT and CaBP1/2-deficient inner hair cells (IHCs) recorded in the extracellular solution containing 2 mM of the respective divalent cation. The two types of inactivation were calculated as depicted. (**C**) VDI and CDI were both significantly enhanced in the IHCs of *Cabp1/2*-DKO animals (*asterisks*; Student's *t*-test, p < 0.0001).

However, it has to be noted that the cells in which train stimuli were applied showed a considerably lower calcium current inactivation upon a 500-ms step depolarization than the average (0.36 (*n* = 5) vs 0.47 (*n* = 21)). Short stimuli at different potentials revealed maximal Ca²⁺-current amplitudes in the IHCs of DKO mice that were comparable to controls (*Figure 1E*). We observed a slight shift in the voltage dependence of Ca$_V$1.3 activation toward more hyperpolarized values (*Figure 1F*; potential of half-activation – $V_{0.5}$ of –33.6 ± 0.5 to –30.9 ± 0.5 mV in *Cabp1/2* DKO and WT IHCs, respectively; p < 0.0001, Student's *t*-test), and an increased slope of the Boltzmann fit function (7.6 ± 0.1 vs 7.1 ± 0.1, p < 0.0001, Student's *t*-test). This finding is somewhat surprising as the opposite would be expected based on data from heterologous expression systems (*Haeseleer et al., 2004*; *Picher, 2017*; *Shaltiel et al., 2012*). At present we cannot exclude potential changes in the relative amounts of different Ca$_V$1.3 splice variants, auxiliary subunits, or other interaction partners that may in the absence of CaBPs lead to these changes in the IHCs.

To further investigate the nature of the enhanced Ca$_V$1.3 inactivation, we recorded Ca$_V$1.3 currents in the presence of 2 mM Ca²⁺ or Ba²⁺ in the extracellular solution (*Figure 2*), and analyzed the two types of inactivation as described previously (*Tadross et al., 2010*). First, VDI was calculated from the Ba²⁺ currents, followed by the calculation of the CDI from the comparison between the Ba²⁺ and Ca²⁺ currents (for mathematical formulas, see Methods). The lack of CaBP1 and 2 significantly enhanced both, VDI and CDI of IHC Ca$_V$1.3 channels, which were approx. two and six times larger than those in WT IHCs (*Figure 2B*; VDI: 0.30 ± 0.02 vs 0.13 ± 0.02 and CDI: 0.65 ± 0.01 vs 0.11 ± 0.02 in *Cabp1/2* DKO and WT IHCs, respectively; p < 0.0001, Student's *t*-test).

## Impaired synaptic transmission in the IHCs lacking CaBP1 and 2

Since Ca²⁺ influx through Ca$_V$1.3 channels triggers IHC neurotransmission we next investigated how the prominent inactivation affects exocytosis in the CaBP1/2-deficient IHCs. Exocytosis was probed by recordings of membrane capacitance (*Figure 3*). In response to short stimuli [primarily recruiting the readily releasable pool (RRP) of vesicles], increases in capacitance were comparable, while long stimuli (reporting sustained exocytosis) evoked significantly smaller responses in the CaBP1/2-deficient IHCs (*Figure 3A, B*). These results are in line with the observation of comparable peak Ca²⁺-current amplitudes but reduced calcium charge transfer ($Q_{Ca}$) for stimulus durations above 50 ms. However, calculating exocytosis efficiency revealed that the reduction in sustained exocytosis cannot completely be explained by the diminished Ca²⁺ influx (*Figure 3C*). This suggests an additional impairment in exo- or endocytic processes, the underlying mechanism of which is currently unknown. Transgenic expression of CaBP2 largely restored the efficiency of exocytosis, indicating that the impairment as observed in the IHCs of DKOs is related to the absence of the CaBPs. Such exocytosis defect was not observed in *Cabp2*-KOs, suggesting that when IHCs are held at hyperpolarized potential at RT, CaBP1 provides sufficient compensation to obscure the synaptic phenotype (*Picher, 2017*; *Oestreicher et al., 2021*).

Next, we have reasoned that holding a cell at a relatively hyperpolarized potential of –85 mV as typically utilized in patch-clamp recordings may not reveal the synaptic phenotype in its severity. We

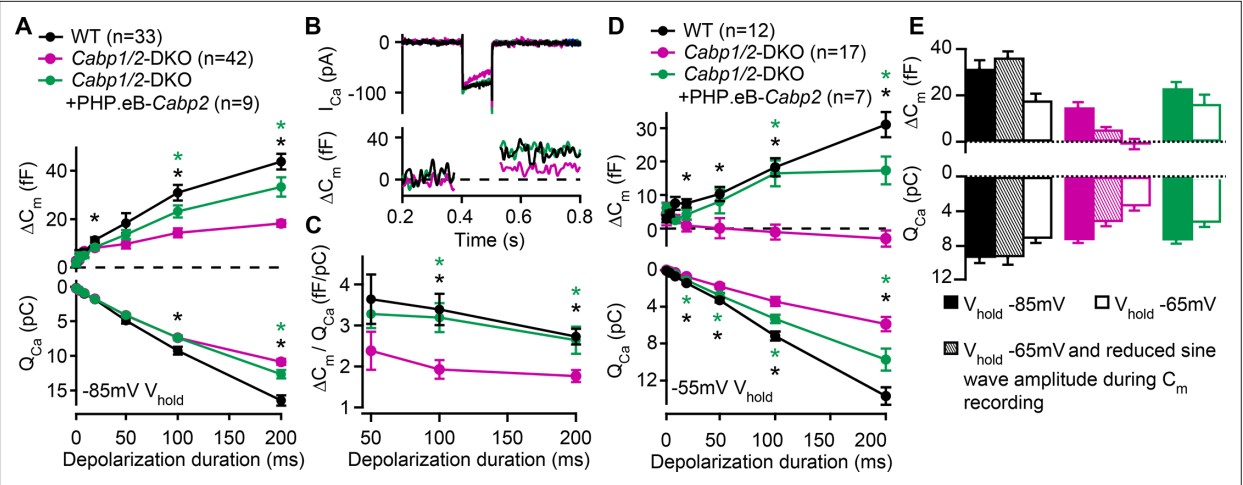

**Figure 3.** Reduced sustained component of exocytosis and calcium charge transfer ($Q_{Ca}$) can be partially rescued by *Cabp2*-transgene expression. (**A**) Capacitance increments with the corresponding $Q_{Ca}$ were probed by different depolarization durations from a holding potential of –85 mV (unpaired *t*-tests with Welch correction). CaBP1/2-deficient inner hair cells (IHCs) showed significant impairment of the sustained exocytosis (p < 0.05 for 20 ms, and <0.001 for 100- and 200-ms step, respectively) and reduced cumulative $Ca^{2+}$ influx (p < 0.005 for the longest two pulses) as compared to WT controls (*black asterisks*). AAV-mediated delivery of *Cabp2* improved IHC synaptic function as compared to non-injected *Cabp1/2*-DKO controls (p < 0.01 for $\Delta C_m$ upon 100- and 200-ms step, and p < 0.05 for $Q_{Ca}$ upon 200-ms step; *green asterisks*). (**B**) Representative $Ca^{2+}$-current traces and corresponding membrane capacitance changes upon 100-ms long depolarization steps to the peak $Ca^{2+}$-current potential. (**C**) Note a reduced efficiency of $Ca^{2+}$-dependent exocytosis (p < 0.005 for the two longest test pulses; *black asterisks*), which can be efficiently rescued by intracochlear delivery of PHP.eB-*Cabp2* (p < 0.01 and 0.05 for 100- and 200-ms pulse, respectively; *green asterisks*). (**D**) Capacitance measurements at a holding potential of –55 mV reveal an aggravation of the phenotype by additional activation of calcium channels between the test pulses (p < 0.01 for 50 ms, and p < 0.005 for 20-, 100-, and 200-ms pulse) and a further discrepancy between the $Q_{Ca}$ of WT and CaBP1/2-deficient IHCs (2–10 ms: p < 0.05; 20–200 ms: p < 0.001). Also in these recording conditions, the IHCs from *Cabp2*-injected *Cabp1/2*-DKO animals showed increased $Q_{Ca}$ (20–200 ms: p < 0.05) and exocytosis (100–200 ms: p < 0.005) as compared to non-injected controls. (**E**) Capacitance increments and the corresponding $Q_{Ca}$ upon 100-ms depolarization steps as recorded in different conditions. Note worsening of the phenotype with increasing IHC activation between the test pulses.

The online version of this article includes the following source data for figure 3:

**Source data 1.** Values obtained from patch-clamp exocytosis experiments that were used in *Figure 3*.

**Source data 2.** Representative traces and values obtained from patch-clamp exocytosis experiments and custom-written macros that were used for analyses in *Figure 3*.

thus performed additional sets of experiments (*Figure 3D, E*), in which the cells were held at either –65 or –55 mV, closer to the suggested resting potential of IHCs (*Russell and Sellick, 1983*; *Dierich et al., 2020*; *Oliver et al., 2003*). Raising the holding voltage to –55 mV resulted in a significant reduction in the maximal $Ca^{2+}$ currents, which was more pronounced in the CaBP1/2-deficient IHCs as compared to controls (60 ± 4% vs 25 ± 4% reduction in DKO and WT IHCs, respectively; p < 0.005, Student's *t*-test). Consequently, further discrepancy between the $Q_{Ca}$ in IHCs of WT and DKO animals was observed (p < 0.001), which in DKO animals resulted in a complete loss of sustained exocytosis (*Figure 3D*; p < 0.005). The latter was well restored upon AAV-mediated transgenic expression of CaBP2 (p < 0.05), which also reduced the amount of calcium channel inactivation (38 ± 8% reduction in the maximal $Ca^{2+}$ current as well as a partial recovery of $Q_{Ca}$ as compared to *Cabp1/2*-DKO IHCs; p < 0.05). Rapid voltage oscillations during capacitance recordings may additionally activate calcium channels prior to the depolarizing step. We thus reduced the sine wave amplitude, which resulted in an intermediate phenotype, indicating pronounced activity-dependent deterioration of synaptic function in the absence of CaBPs (*Figure 3E*). We propose that in vivo in the absence of CaBP1 and 2 a large fraction of IHC $Ca_V1.3$ channels are in a state of prolonged inactivation. This does not support sufficient RRP exocytosis (*Figure 3D*) to sustain sensory coding during ongoing stimulation of IHCs.

Lastly, we asked whether the deletion of CaBP1 and 2 affects the speed of recovery from $Ca_V1.3$ inactivation (*Figure 4*). We applied an initial 5-ms long pulse followed by a conditioning stimulus and a series of 5-ms long pulses to test the kinetics of recovery of the $Ca^{2+}$-current amplitude. Upon 500-ms depolarization step (*Figure 4B*), the speed of recovery from inactivation was fast both in DKO as well as WT IHCs where a much smaller fraction of channels underwent inactivation (time

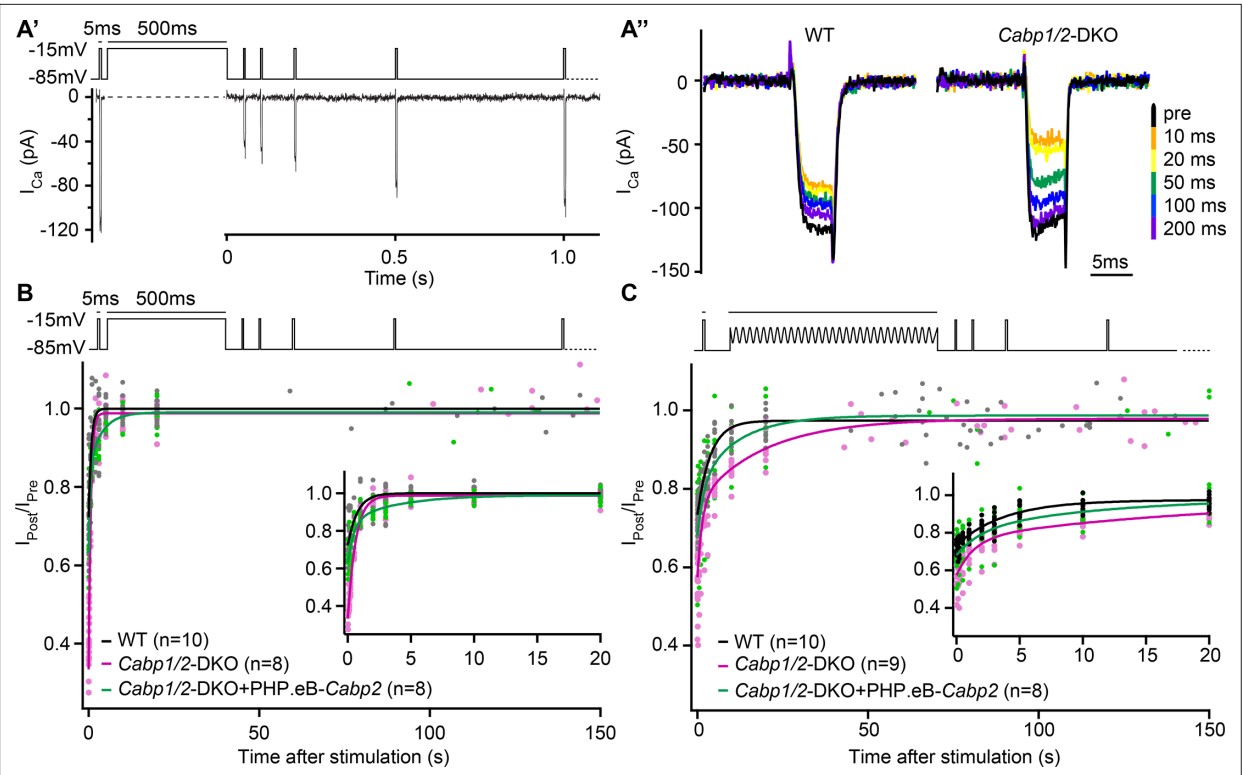

**Figure 4.** The speed of recovery of Ca²⁺ currents is affected in CaBP1/2-deficient inner hair cells (IHCs). (**A**) Ca²⁺ currents were probed by 5-ms long depolarization steps to the maximum current potential. An initial test pulse was followed by a conditioning stimulus (a 500-ms depolarization step or a 120-s long sine wave). Subsequently, Ca²⁺ currents were tested at various time points to assess the recovery of Ca²⁺-current amplitudes. (**A'**) Example Ca²⁺-current recording using a 500-ms long depolarization step as a conditioning stimulus. (**A''**) Representative Ca²⁺-current traces in WT and CaBP1/2-deficient IHCs at different time points after a 500-ms long conditioning stimulus. Ca²⁺-current amplitudes normalized to the amplitude of the pre-conditioning test pulse up to 150 s after a 500-ms depolarization step (**B**) or a 2-min long sine wave stimulus (**C**). The inserts show the initial 20 s after the conditioning stimulus. The data were fitted by a single or double exponential function (fit parameters in *Table 1*). Note stronger inactivation in the *Cabp1/2*-DKO IHCs and delayed recovery of Ca²⁺ currents after the presentation of a long sine wave and a partial recovery upon viral gene transfer of *Cabp2*. Recovery upon a 500-ms pulse compared to the 2-min long sine wave stimulus was much faster. Surprisingly, in these conditions recovery of calcium currents was slowest in hair cells with virus-mediated re-expression of CaBP2.

The online version of this article includes the following source data for figure 4:

**Source data 1.** Traces and values obtained from patch-clamp recovery experiments and custom-written macros that were used for analyses in *Figure 4*.

constants of 0.62 and 0.75 s in DKO and WT IHCs, respectively; *Table 1*, left). As expected, transgenic expression of CaBP2 reduced the initial inactivation (Ca²⁺-current amplitude 10 ms after the conditioning pulse, expressed as the fraction of initial current – $I_{10ms\_post}/I_{pre}$, WT: 0.73 ± 0.01; DKO: 0.36 ± 0.03; DKO + PHP.eB-*Cabp2*: 0.64 ± 0.02; p < 0.001 for DKO vs DKO + PHP.eB-*Cabp2*). However, the recovery from the step-evoked inactivation was slowest in the *Cabp2*-injected DKO IHCs. We hypothesize that this may be related to the speed at which the intracellular [Ca²⁺] after depolarization is restored to the basal level allowing Caᵥ1.3 recovery from CDI (see discussion). A 2-min long sine wave (±15 mV) from –55 mV to mimic a milder but prolonged hair cell stimulation (*Figure 4C*) elicited smaller inactivation of Caᵥ1.3 channels as compared to a 500-ms depolarization step ($I_{10ms\_post}/I_{pre}$ of 0.73 ± 0.01 in WT; 0.58 ± 0.03 in DKO; 0.68 ± 0.03 in DKO + PHP.eB-*Cabp2*; p < 0.05 for DKO vs DKO + PHP.eB-*Cabp2*). In these conditions, however, the recovery from inactivation was slowest in the DKO IHCs (*Table 1*, right). This suggests that in the absence of CaBP1 and 2 even mild but prolonged stimulation of IHCs in vivo may drive a significant fraction of Caᵥ1.3 channels into 'deep-state' of inactivation with very slow recovery, as was described for Naᵥ, Kᵥ, and Caᵥ channels (*Bezanilla et al., 1982*; *Payandeh et al., 2012*; *Olcese et al., 1997*; *Ferreira et al., 2003*), not allowing efficient sound encoding.

**Table 1.** Ca$^{2+}$-current recovery fit parameters.

The data was fitted with the single or the double exponential with the following equations: $y(t) = y_0 - A_1 \times \exp(-(t/\tau_1))$ and $y(t) = y_0 - A_1 \times \exp(-(t/\tau_1)) - A_2 \times \exp(-(t/\tau_2))$, where $y(t)$ represents the peak current amplitude at the time $t$ after the end of conditioning pulse (either 500-ms depolarization or 2-min sine wave) given in % of the initial pre-conditioning peak current amplitude ($100 \times I_{post}/I_{pre}$). While current recovery upon 500-ms depolarization in the inner hair cells (IHCs) of WT and DKO animals could be well fitted by a single exponential, the data obtained in the injected DKO animals required a double exponential fitting. Upon 2-min sine wave, the current recovery in DKO IHCs (both, injected and non-injected group) also required fitting with a double exponential function.

| Test group | 500-ms depolarization | | | | | 2-min sine wave | | | | |
|---|---|---|---|---|---|---|---|---|---|---|
| | $A_1$ (%) | $\tau_1$ (s) | $A_2$ (%) | $\tau_2$ (s) | $y_0$ (%) | $A_1$ (%) | $\tau_1$ (s) | $A_2$ (%) | $\tau_2$ (s) | $y_0$ (%) |
| DKO | 65 | 0.62 | - | - | 99 | 19 | 1.36 | 21 | 19.41 | 98 |
| WT | 28 | 0.75 | - | - | 100 | 24 | 3.83 | - | - | 97 |
| DKO + PHP.eB-*Cabp2* | 22 | 0.47 | 14 | 3.87 | 99 | 13 | 1.88 | 18 | 11.56 | 99 |

## Altered synaptic morphology in *Cabp1/2*-DKO mice

Immunolabeling of the organs of Corti indicated normal presynaptic Ca$_V$1.3 clustering with no apparent change in size or shape of the clusters (*Figures 5A*), and comparable abundance of ribbon synapses in the apical IHCs of WT and DKO animals (12.9 ± 0.2 vs 12.8 ± 0.8, respectively; *Figure 5B*) at 3 weeks of age. However, the intensities of pre- (RibeyeA/CtBP2) and postsynaptic (Homer1) immunofluorescent puncta were increased in the CaBP1/2-deficient IHCs as compared to age-matched controls, processed in parallel (*Figure 5G*). A similar alteration in the synaptic size was observed in some other synaptic mutants as well as upon noise trauma (*Eckrich et al., 2019*; *Kim et al., 2019*), but not in *Cabp2* single KOs (*Picher, 2017*), which show significantly less impaired presynaptic Ca$^{2+}$ signaling. This data is in line with compensatory postsynaptic changes in conditions of no or very little postsynaptic activity, possibly governed through adaptive changes in the postsynaptic elements.

As earlier studies of Ca$_V$1.3 channelopathies demonstrated a lag or stop in IHC development (*Eckrich et al., 2019*; *Brandt et al., 2003*; *Neef et al., 2009*) we next examined the expression of small- (SK) and large-conductance Ca$^{2+}$-activated potassium channels (BK) in the IHCs at different ages (*Figure 5C–F*). As expected for normal maturation, the SK2 puncta were mostly absent from DKO IHCs by 3 weeks of age and the BK puncta underwent a typical transition from a spotty appearance throughout the cell membrane to the patches at the IHC necks (*Figure 5F*). However, by 3 weeks and more obviously later on, the BK channel clusters appeared smaller in the DKO IHCs as compared to controls, which could be partially reversed by virally mediated expression of CaBP2 (*Figure 5E*). Using Imaris, we analyzed the respective 'volumes' of BK-cluster immunofluorescence in the IHC neck region from ca. 4-week-old mice. This analysis revealed significantly smaller (*Figure 5H*; Student's *t*-test, p < 0.0001, N = 6 organs per genotype) and fewer clusters in DKO IHCs. This was also obvious when comparing the cumulative 'volume' of BK-cluster immunofluorescence per IHC, which was significantly higher in WT IHCs, indicating higher abundance of BK channels (14.2 ± 2.1 vs 6.6 ± 1.1 µm$^3$ per IHC, Student's *t*-test, p < 0.01). We hypothesize that CaBPs may directly (e.g. as transcription factors) regulate BK expression, a notion that needs to be tested in the future. Alternatively, the effects may be more indirect, as they are common in mutants with impaired intracellular Ca$^{2+}$ signaling (*Eckrich et al., 2019*; *Brandt et al., 2003*; *Neef et al., 2009*).

## Severe hearing impairment in mice deficient in CaBP1 and CaBP2

Hearing was assessed by measuring auditory brainstem responses (ABRs; *Figure 6*, *Figure 6—figure supplement 1*). Already in young (3- to 4-week-old) *Cabp1/2*-DKO animals, ABR thresholds were highly elevated, with tone bursts up to a maximum intensity of 90 dB SPL barely eliciting any ABR response. Click responses were better preserved (thresholds of 59 ± 2 dB SPL in DKO (N = 9) vs 30 ± 2 dB SPL in WT (N = 8) mice) but greatly reduced in amplitude (*Figure 6—figure supplement 1A, B*). At 7–13 weeks of age, further worsening of the click thresholds was observed (68 ± 5 dB SPL in DKO

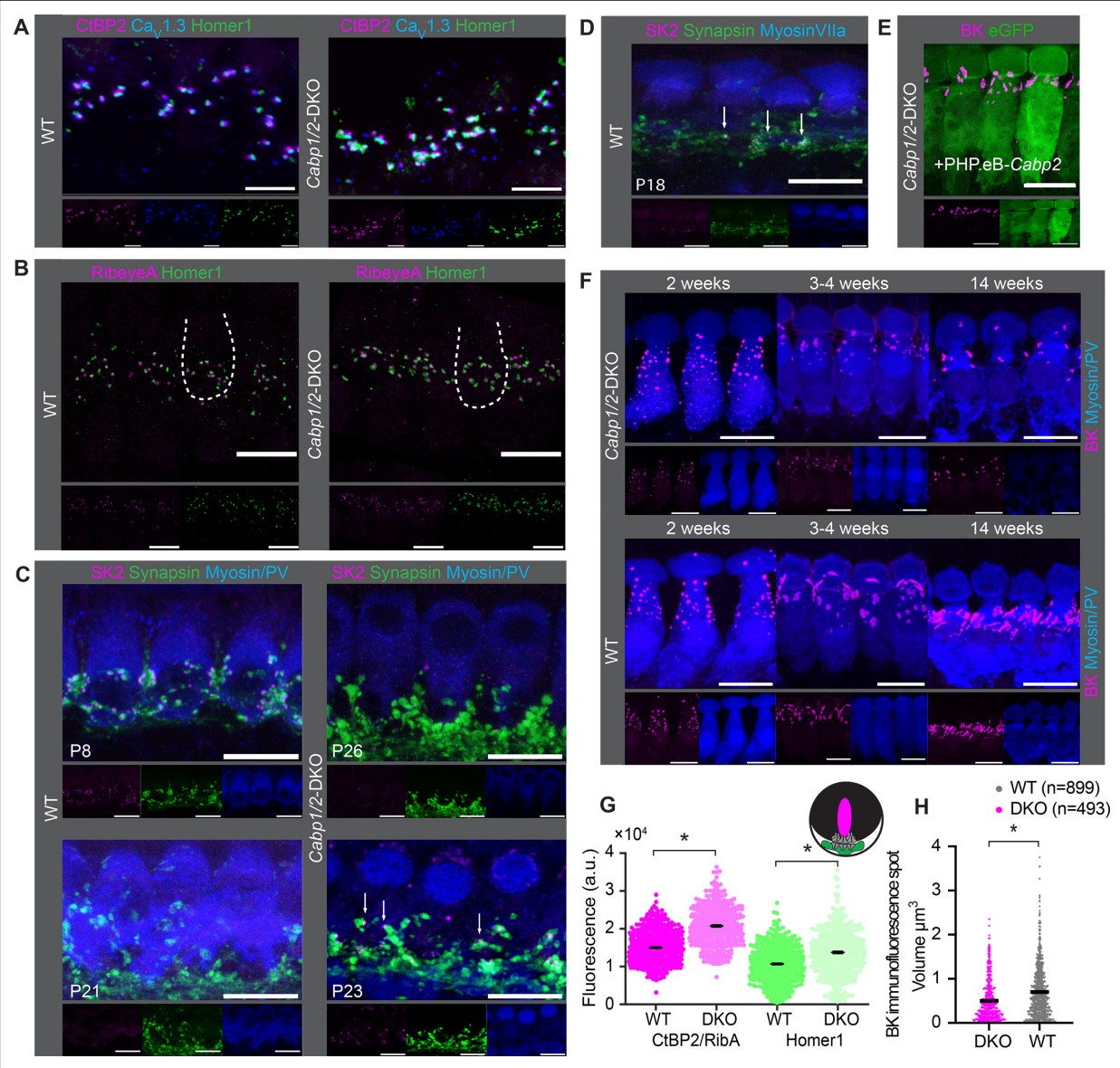

**Figure 5.** Comparable synapse density with changed morphology in *Cabp1/2*-DKO inner hair cells (IHCs). (**A**) Ca$_V$1.3-immunofluorescence appeared normal in *Cabp1/2*-DKO-mice. Scale bars: 5 μm. (**B**) Ribbon synapse immunostainings using RibeyeA as a presynaptic and Homer1 as a postsynaptic marker. Ribbon synapses were identified as juxtaposed pre- (RibeyeA/CtBP2) and postsynaptic (Homer1 or, in some organs, GluA3) immunofluorescent spots. (**C**) SK2-immunofluorescence was absent from the majority of IHCs in 3-week-old animals, but prominent in immature cochleae of 1-week-old animals and together with synapsin 1 labeled direct synaptic contacts from the efferent fibers to the IHCs. Occasionally, however rarely, isolated SK2 immunofluorescent spots were still detected in close proximity to the synapsin marker in the IHCs of 3-week-old DKO animals (*arrows*). (**D**) Expression of SK2 beyond the second postnatal week was also observed in some WT IHCs (*arrows*). (**E, F**) BK clusters at 3 weeks of age appeared smaller in the non-treated animals, and less so in the ears of PHP.eB-*Cabp2*-injected *Cabp1/2*-DKO animals. This difference became even stronger at 14 weeks of age. Scale bars are 10 μm if not stated otherwise. Note that myosin VIIa or parvalbumin-alpha (PV) were used as IHC markers. (**G**) A significantly higher immunofluorescence intensity was observed for pre- (RibeyeA or CtBP2) and postsynaptic (Homer1) markers in *Cabp1/2*-DKO mice (analysis of 614 and 441 immunospots from WT and *Cabp1/2*-DKO organs of Corti, respectively; *N* = 5 for both genotypes; *asterisks*; Wilcoxon rank-sum test, p < 0.005). (**H**) The volume of BK-channel immunofluorescent spots around the neck of the IHCs was significantly smaller in DKO IHCs (*asterisk*, Student's *t*-test p < 0.0001, *N* = 6 organs per genotype).

The online version of this article includes the following source data for figure 5:

**Source data 1.** Values obtained from inner hair cell (IHC) ribbon synapse images that were used for analyses in *Figure 5G*.

**Source data 2.** Values obtained from inner hair cell (IHC) BK channel images that were used for analyses in *Figure 5H*.

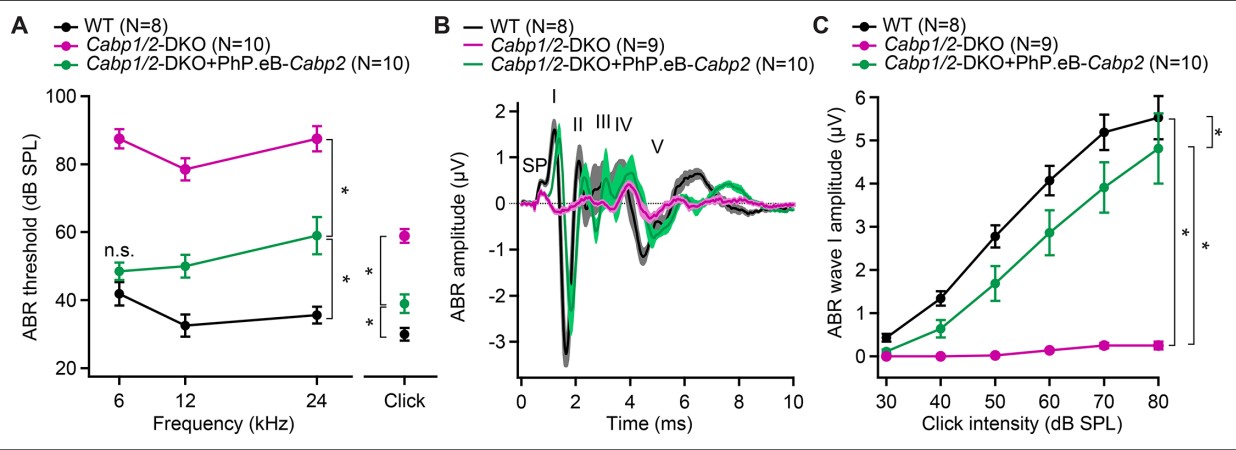

**Figure 6.** AAV-mediated transgene-expression of *Cabp2* ameliorates hearing loss and recovers auditory brainstem response (ABR) amplitude responses in *Cabp1/2*-DKO mice. (**A**) Significant improvement of ABR thresholds in the injected ears of 3- to 4-week-old *Cabp1/2*-DKO animals for all tone burst frequencies tested as well as 20-Hz-click stimulus (*asterisks*, see Table in **Supplementary file 1** for the p values). (**B**) Average click-evoked responses to an 80-dB 20-Hz-click stimulus. (**C**) A strong recovery of the amplitude of the ABR wave I (*asterisks*, Tukey's multicomparison test; p ≤ 0.0001 for DKO vs WT or vs injected DKO) upon *Cabp2*-transgene expression.

The online version of this article includes the following source data, source code, and figure supplement(s) for figure 6:

**Source data 1.** Values obtained from auditory brainstem response (ABR) recordings that were used for analyses in *Figure 6A–C*, *Figure 6—figure supplement 1A, B*, and *Figure 6—figure supplement 2C, D*.

**Figure supplement 1.** Auditory brainstem response (ABR) amplitudes are strongly decreased while distortion product otoacoustic emission (DPOAE) are mostly preserved in the *Cabp1/2*-DKO animals.

**Figure supplement 1—source code 1.** Matlab codes used to analyse DPOAE data.

**Figure supplement 1—source data 1.** Values obtained from DPOAE recordings that were used for analyses in *Figure 6—figure supplement 1C and E*.

**Figure supplement 1—source data 2.** Values obtained from DPOAE recordings that were used for analyses in *Figure 6—figure supplement 1D and F*.

**Figure supplement 2.** Efficient transgene expression of *Cabp2* recovers cumulative auditory brainstem response (ABR) amplitude responses and ABR wave I latencies in *Cabp1/2*-DKO mice.

**Figure supplement 2—source code 1.** Matlab source code for GFP fluorescence analysis.

**Figure supplement 2—source data 1.** Values obtained from eGFP immunostainings that were used for analyses in *Figure 6—figure supplement 2B*.

vs 30 ± 1 dB SPL in WT mice; *N* = 7 for both). Upon 80 dB SPL click stimulation, no sizable ABR wave I was detected in average traces of DKO animals at either age tested (*Figure 6—figure supplement 1B*), suggesting reduced and/or desynchronized spiking of spiral ganglion neurons (SGNs). Like in other mouse mutants with defects in the IHC synapse (*Khimich et al., 2005*; *Strenzke et al., 2016*), the later ABR peaks were better preserved, indicating partially preserved activation of neurons of the auditory brainstem. To assess cochlear amplification by the outer hair cells, we recorded distortion product otoacoustic emissions (DPOAEs). While OAEs were so far only detectable in some of the younger DFNB93 patients (*Picher, 2017*; *Bharadwaj et al., 2014*), mouse work corroborates the notion of at least initially intact cochlear amplification in the absence of CaBP2 (*Yang et al., 2018*; *Picher, 2017*; *Oestreicher et al., 2021*). Our results suggest that cochlear amplification is largely preserved also in DKO animals (*Figure 6—figure supplement 1C–F*). In about half of the DKO animals in both age groups, the DPOAEs were normal. In the other half, they were reduced in amplitude but still detectable (*Figure 6—figure supplement 1E, F*), which may suggest partially hampered cochlear amplification in a subset of DKO animals.

Transgenic expression of CaBP2 led to significant improvement of the IHC Ca$_V$1.3 function and synaptic exocytosis (*Figures 1–4*), we thus next asked, how this may influence the hearing of the animals. As previously observed (*Oestreicher et al., 2021*), unilateral injection of the PHP.eB-*Cabp2*-eGFP resulted in strong eGFP immunofluorescence signal in the vast majority of hair cells and elsewhere (*Figure 6—figure supplement 2A, B, E*). Transgenic expression of CaBP2 in DKO animals led

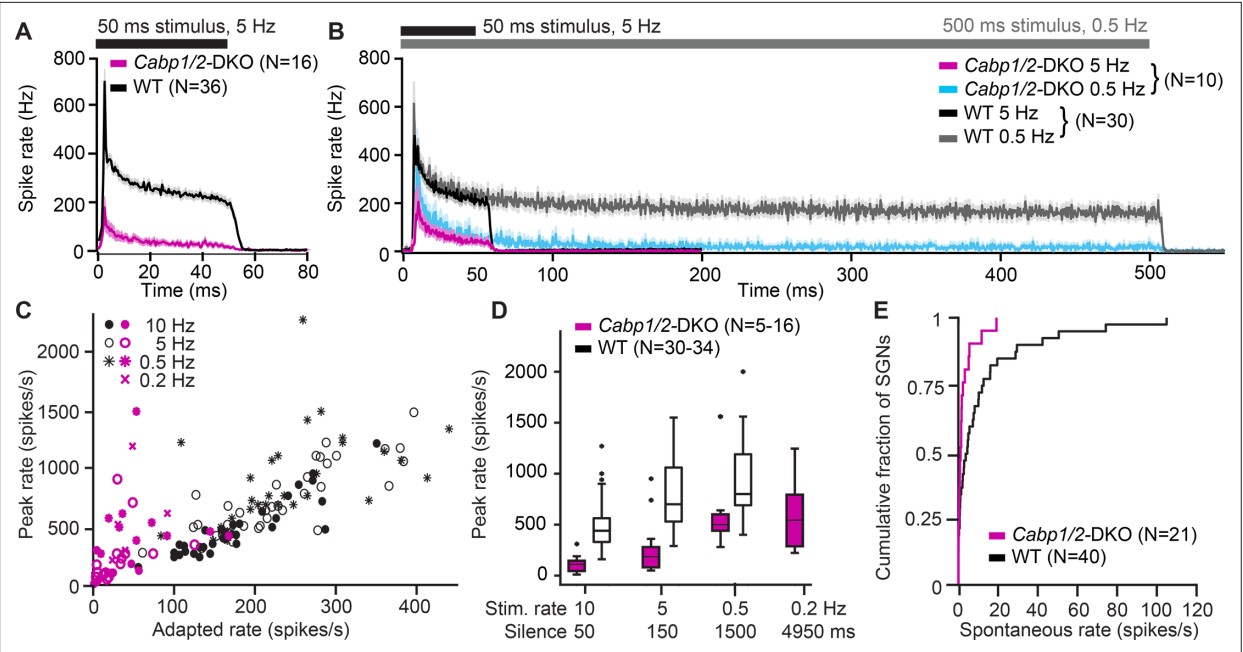

**Figure 7.** Impaired sound encoding in *Cabp1/2*-DKO mice. Single neuron responses to sound stimulation in vivo in the region of the auditory nerve of *Cabp1/2*-DKO mice were extremely scarce. (**A**) WT poststimulus time histograms of spiral ganglion neurons (SGNs) to 50 ms suprathreshold tone or noise burst stimuli presented at a rate of 5 Hz showed a very high sound onset rate gradually adapting to a sustained rate persisting throughout the stimulus duration. Spike rates were much lower in the mutants. (**B**) Applying a lower stimulus rate of 0.5 Hz partially restored sound onset responses in *Cabp1/2*-DKOs, highlighting the enhanced strength of adaptation. WT and mutant neurons sustained their adapted spiking response throughout the longer 500 ms stimulus used in these recordings. For direct comparison, the average 5 Hz poststimulus time histograms from the same neurons are shown. (**C**) Correlations of peak and adapted rates for different stimulus rates. (**D**) Peak rates for different stimulus rates/interstimulus intervals in WT SGNs strongly increase between 10 and 5 Hz, while in DKOs a longer period of silence is required to observe such difference. (**E**) *Cabp1/2*-DKO neurons had lower spontaneous spike rates than WT SGNs.

The online version of this article includes the following source code for figure 7:

**Source code 1.** Matlab source codes for analysis of SU data.

to significantly improved hearing with lower ABR thresholds, increased click-evoked ABR amplitudes, and shortened wave I latency (***Figure 6***, ***Figure 6—figure supplement 2C, D***).

Finally, we studied sound encoding by inspecting sound-induced spiking responses of SGNs (***Figure 7***). In WT mice, we recorded responses in 39 neurons, which were classified as SGNs based on their location, response patterns, spike waveforms and recording stability. Their spiking characteristics resembled typical WT data (***Picher, 2017***; ***Jean et al., 2018***; ***Jing et al., 2013***; ***Taberner and Liberman, 2005***; ***Vogl et al., 2017***). In response to 50- or 500-ms tone burst at the characteristic frequency 30 dB above threshold (i.e. where the sigmoidal rate-intensity functions have saturated spike rates) and at a stimulus repetition rate of 5 or 0.5 Hz, they showed the SGN-typical very high onset firing rates, gradually adapting to reach steady-state rates between 100 and 400 Hz (***Figure 7A, B***). In contrast, only one out of the 20 neurons we recorded and analyzed from DKO animals showed a normal response in our automated tuning curve algorithm with a threshold of 35 dB SPL at 7.8 kHz, while all others displayed thresholds above 80 dB SPL and extremely low spike rates (***Figure 7A, B***). Since in the majority of neurons frequency tuning was very broad, we used subjectively chosen best frequency tone burst or white noise burst stimulation of at least 80 dB SPL for subsequent characterization. 16 out of 19 DKO units responded with a relatively high onset rate but strong adaptation to a minimal sustained spiking (***Figure 7A, B***), which prevented unit classification based on the response patterns. Judging from the criteria described above, we assume that approximately 13 of these represent SGNs and 3 represent bushy cells or multipolar cells of the anteroventral cochlear nucleus. Three additional units (two resembling a pauser/buildup- and one displaying a clear chopper neuron-type response) were excluded from further analysis. Overall, in DKO neurons, peak and adapted rates were greatly reduced as compared to WT controls (***Figure 7C***), which came along with an increased latency

(12.5 ± 2.8 ms vs WT 4.8 ± 0.2 ms; p < 0.001, Wilcoxon rank-sum test) and increased jitter (18.3 ± 5.0 vs WT 1.5 ± 0.4; p < 0.00001, Wilcoxon rank-sum test) of the first spike that followed the sound onset.

DKO responses greatly improved when we lowered the stimulus repetition rate to 0.5 Hz, and plateaued at 0.2 Hz (*Figure 7C, D*). We hypothesize that in DKO SGNs, recovery of $Ca_V1.3$ channels after a silent interval in the range of 1–2 s is the limiting factor for determining the maximal SGN spiking rate at the sound onset. This may be further contributed by potentially slowed re-supply of synaptic vesicles, as suggested by reduced efficiency of sustained exocytosis in the CaBP1/2-deficient IHCs (see *Figure 3*). In contrast, in WT SGNs, the maximal spike rates are mostly recovered after a silent interval of 150 ms, which may be mostly limited by vesicle replenishment at the IHC ribbon synapse (*Strenzke et al., 2016*; *Pangrsic et al., 2010*). During a 5-Hz sound stimulation, the strength of adaptation was greatly enhanced (peak to adapted ratio 12.5 ± 3.0 vs WT 3.4 ± 0.2; p < 0.00001, Wilcoxon rank-sum test) whereas the time constant of adaptation was unchanged 7.9 ± 2.0 ms in DKO compared to 6.9 ± 0.4 ms in WT (p = 0.5, Wilcoxon rank-sum test). However, for the longer 500 ms tone bursts used in the 0.5 Hz stimulation dataset, it becomes obvious that despite the strong adaptation, DKO neurons are still able to sustain spiking at a low rate throughout the stimulation (*Figure 7B*). DKOs also showed reduced spontaneous spiking activity (*Figure 7E*), which supports the hypothesis that already in the absence of sound stimulation over tens of seconds a considerable fraction of calcium channels may be in steady-state inactivation.

The reduction in spike rates and their strong dependence on the strength of stimulation in the preceding seconds introduces experimental bias to our data. In the units of the DKO animals, it was not feasible to faithfully assess sound thresholds or dynamic ranges. Furthermore, the overall experimental yield was generally much lower in DKO animals. On average we hit 0.3 sound-responsive neurons per hour of recording, compared to 2.5 in WT. Very likely many DKO neurons failed to generate spontaneous and evoked action potentials, thus we were not able to identify them. Based on the DPOAE data we assume that cochlear mechanics and active amplification and thus the IHC receptor potential thresholds are mostly normal in the DKOs. We cannot exclude that IHC receptor potentials are decreased in amplitude, which however should rather reduce SGN adaptation. We thus interpret the observed increase in adaptation and delay of recovery of SGN spiking to be the effects of changed synaptic $Ca_V1.3$ channel kinetics as described above.

In summary, sound responses of DKO SGNs and cochlear nucleus neurons showed a drastic reduction in spontaneous and evoked action potential rates, which was associated with increased sound thresholds and reduced temporal precision of action potential generation. Together, these factors explain well the poor ABR thresholds (*Figure 6*, *Figure 6—figure supplement 1A*).

## Discussion

In this study, we examined the effects of genetic disruption of the two major CaBPs expressed in the cochlea. In the IHCs of CaBP1/2-deficient animals, we found largely inactivated $Ca^{2+}$ currents with prolonged recovery from inactivation upon mild oscillatory stimulation, which did not support sufficient neurotransmission for proper sound encoding. The extent of channel inactivation and impairment of IHC synaptic exocytosis aggravated with the extent of IHC stimulation. Similarly, SGN action potential rates in vivo declined at increased sound stimulation rates and displayed very strong adaptation upon stimulation, going along with impaired sustained IHC exocytosis. The impairment was significantly stronger as compared to the phenotype in the absence of CaBP2 alone. This suggests that under in vivo conditions the roles of CaBP1 and 2 are partially overlapping.

### Function and expression profile of CaBPs in the cochlea

The murine cochlea shows strong expression of CaBP1 and 2, while the levels of CaBP4 and 5 seem negligible (*Yang et al., 2016*; *Jean et al., 2023*; *Xu et al., 2022*). Among the three putative splice isoforms of CaBP2 differing in the N-terminal aminoacid sequence, the alternative long isoform, termed CaBP2-alt, may be the most abundant splice variant in the IHCs, as suggested by in situ hybridization (*Yang et al., 2016*), while SGNs seem to express none. So far our gene-rescue experiments were conducted with the conventional L-isoform only, which however shows strong inhibition of $Ca_V1.3$ inactivation in the HEK cells as well as IHCs (*Picher, 2017*; *Oestreicher et al., 2021*).

Alternative splicing also generates three CaBP1 variants (*Yang et al., 2016*). The alternative long variant caldendrin is primarily expressed in the SGNs or the surrounding glia cells, while weak expression of CaBP1-S/-L (as judged by the mRNA production) was detected in hair cells (*Yang et al., 2016*). Our DKO model lacks all splice isoforms of both CaBPs. Since *Cabp1* KO animals show normal hearing several weeks after birth, and CaBP2 is absent from SGNs, we interpret the hearing impairment of the *Cabp1/2* DKO animals to be primarily due to defects of hair cell function and judge contribution by SGN dysfunction as rather unlikely, at least in the early postnatal weeks.

The ABR thresholds of *Cabp1* KO animals are normal at 4 weeks of age, but deteriorate with age, starting with high-frequency hearing loss at 9 weeks. Conversely, the suprathreshold ABRs to tone burst stimulation show increased amplitudes and reduced latencies during the first 2 months after birth. This suggested enhanced SGN activity at first, followed by slowly progressing functional decline, possibly due to overexcitability-related toxicity (*Yang et al., 2018*). Due to early strong impairment of hair cell synaptic function, the latter effects are not expected to occur in the DKO animals. However, ABR thresholds for clicks were further increased and an additional reduction in the summating potential was observed in the DKO animals at two as compared to 1 month of age despite relatively well-preserved DPOAEs (*Figure 6—figure supplement 1A–D*). It thus remains to be clarified whether the absence of CaBP1 and 2 may affect (IHC) mechanotransduction at more mature stage. Unfortunately, this is difficult to assess due to the increased fragility of hair bundles at mature age. Intriguingly, CaBPs share similarities with calcium and integrin-binding protein 2 (CIB2), associated with hair bundle defects (*Giese et al., 2017*; *Riazuddin et al., 2012*; *Wang et al., 2017*). When we overexpressed CaBP2 in the DKO animals, interestingly, suprathreshold responses of ABR wave I were very large (*Figure 6*) and the summated wave I–IV responses were not distinguishable from WT responses (*Figure 6—figure supplement 2C*). However, tone burst and click thresholds did not completely recover to WT levels. The difference in the extent of rescue of threshold and suprathreshold responses may be due to possibly distinct requirements for CaBPs at synapses providing inputs to low- or high-threshold SGN fibers. Apart from a small shift in voltage activation and abundance of the channels (*Ohn et al., 2016*), not much is known about the possible differences in the $Ca_V1.3$ properties and interaction partners at distinct ribbons. Neither is it known whether the composition of the channel complex t.i. proportions of alpha subunit splice variants (*Scharinger et al., 2015*; *Singh et al., 2008*; *Vincent et al., 2017*), and different beta subunits (*Ortner et al., 2020*) differs systematically between ribbon-AZs partnered by different classes of SGN fibers. Investigation of intracellular $Ca^{2+}$ signals from individual AZs may offer some answers in the future. Different amounts of rescue of threshold vs suprathreshold responses could be further due to confounding effects of CaBP1 and 2, for example augmented SGN response in *Cabp2*-injected DKO animals still lacking CaBP1 (reminiscent of the phenotype of young *Cabp1* KOs) in combination with 'only' partial rescue of CaBP2 function. Possible adverse effects of exogenous expression of CaBP2, for example off-target toxicity and uncontrolled overexpression of the transgene, have been discussed previously (*Oestreicher et al., 2021*), and further experiments need to be performed in the future to achieve (close to) complete rescue of hearing function in these mutants.

Our results reveal that the amount of $Ca_V1.3$ inactivation increases dramatically when IHCs lack both major CaBPs. This suggests that the sole loss of either CaBP1 or 2, can be largely compensated by the other of the two CaBP members. Yet, the loss of CaBP2 alone still results in progressive hearing impairment in mice and DFNB93 in humans. The inability of CaBP1 to fully replace CaBP2 in the CaBP2-deficient IHCs may stem from either too low concentration of CaBP1 (and other CaBPs), or from distinct properties of the two proteins (enabling CaBP2 to exert more potent suppression of $Ca_V1.3$ inactivation, e.g. VDI, or potentially performing additional, yet unidentified cellular function). Interestingly, even the overexpression of CaBP2 did not completely rescue the hearing phenotype of DKO animals. Future experiments involving overexpression of CaBP1 (and possibly CaBP1-2 molecular chimeras) in *Cabp2* SKO and/or *Cabp1/2* DKO animals together with further biophysical interaction studies are required to gain a better understanding of the potentially distinct functions performed by the two major CaBPs in the hair cells.

## CaBPs as the regulators of IHC synaptic transmission

Previously, patch-clamp recordings surprisingly revealed normal whole-cell exocytosis in CaBP2-deficient IHCs despite enhanced inactivation of calcium channels (*Picher, 2017*), and no obvious

defects in the CaBP1-deficient IHCs (*Yang et al., 2018*). In contrast, IHCs deficient in both, CaBP1 and 2, displayed strongly impaired exocytosis. Transgenic overexpression of CaBP2 in IHCs resulted in a partial recovery of the Ca²⁺ influx and an even better recovery of exocytosis. Together, these results corroborate the hypothesis of the role of CaBPs in preventing extensive steady-state inactivation of $Ca_V1.3$ channels to support sufficient synaptic exocytosis for proper sound encoding (*Picher, 2017*). The lack of CaBP1 and 2 seems to impair sustained exocytosis and efficiency of exocytosis beyond the levels that could be attributed to reduced $Q_{Ca}$. This became particularly apparent when the IHCs were challenged by higher holding potentials in combination with large sine wave amplitudes during capacitance measurements, which we assume to partially deplete the pool of RRP vesicles. At present, it is not clear how CaBPs may directly affect the exocytosis, for example by influencing the coupling of the $Ca_V1.3$ channels and synaptic vesicles or potentially affecting the expression levels of other synaptic proteins that mediate synaptic vesicle replenishment. In this respect, it is worth noting that CaBP1 and 2 have been reported to modulate CaMKII activity (*Haeseleer et al., 2000*), which in turn may through phosphorylation modify the Ca²⁺ affinity of otoferlin C2 domains (*Meese et al., 2017*). In the future, it is thus essential to investigate if CaBPs regulate the function of otoferlin and thus synaptic processes. Furthermore, other members of the CaBP/CaM family have been suggested to interact with synaptic proteins and/or be involved in the regulation of synaptic vesicle cycle: CaM in endocytosis (*Wu et al., 2009*), CaBP5 in priming via interaction with Munc18-1 (*Sokal and Haeseleer, 2011*), CaBP1 via inhibition of IP₃ receptors (*Haynes et al., 2004*; *Kasri et al., 2004*). An important challenge for future studies is to identify further CaBP(2)-interaction partners and possible functions beyond modulation of calcium channels.

The recovery of $Ca_V1.3$ channels from inactivation upon prolonged milder stimulation (2-min sine wave) was following a single exponential course in WT IHCs, but additional slower component was detected in the IHCs deficient in CaBP1 and 2. In the IHCs of injected DKO animals, this second component appeared smaller and faster, suggesting that CaBPs speed-up recovery of $Ca_V1.3$ channels from inactivation. Upon a comparably shorter (500 ms) but stronger depolarization, the recovery of $Ca_V1.3$ from inactivation was very fast in DKO IHCs, in fact faster than in (WT or) *Cabp2*-injected DKO IHCs. We hypothesize this to be due to faster Ca²⁺ clearance from the DKO IHC synapses as the intracellular [Ca²⁺] levels are expected to rise significantly less in CaBP1/2-deficient IHCs due to fast and pronounced $Ca_V1.3$ inactivation. In contrast, higher [Ca²⁺] may buildup at the synapses of WT and 'rescued' DKO IHCs; consequently, CDI may be prolonged. Another plausible explanation for the observation is, if we consider that CaPB2 may act as a slow mobile calcium buffer (*Picher, 2017*). In the presence of excessive 'overexpressed' CaBP2 in injected IHCs of DKO animals, the depolarization-evoked intracellular [Ca²⁺] transients may become biexponential as has been described for parvalbumin alpha (*Lee et al., 2000*), which may prolong the time constant of Ca²⁺ clearance and determine the kinetics of channel recovery from inactivation. Our results of the recovery upon 500-ms depolarization may thus mostly reflect the kinetics of the Ca²⁺ clearance from the cytosol, while the experiments with prolonged sine wave stimulation (when e.g. the function of calcium pumps and other extrusion systems and intracellular [Ca²⁺] levels may be at equilibrium), may report the speed of calcium channel recovery from a slow inactivation. Slow inactivation with very slow recovery was described for $Na_V$, $K_V$, and $Ca_V$ channels (*Bezanilla et al., 1982*; *Payandeh et al., 2012*; *Olcese et al., 1997*; *Ferreira et al., 2003*). It has to be noted that the time constant of recovery of $Ca_V1.2$ from slow VDI was comparably shorter (4 s) as observed in the present study (*Ferreira et al., 2003*). Possibly different conditions (e.g. stimulus parameters) or channel subtype may underlie this difference. Future experiments combining patch-clamp recordings with calcium imaging should be employed to address this question further. Finally, while we consider the effects of surgery on $Ca_V1.3$ recovery kinetics in injected mice as unlikely, we can at present not exclude this possibility.

From our in vitro recordings at higher holding potential or upon prolonged stimulation with small voltage oscillations we predict that in vivo a noteworthy proportion of $Ca_V1.3$ channels may be effectively inactivated and not available for mediating IHC synaptic transmission. This could explain the low experimental yield, the low 'spontaneous' spiking rates of the SGNs and lower sound-evoked spike rates with higher thresholds. Similarly, thresholds may be high for tone bursts in the ABR measurements due to high proportion of inactivated channels upon stimuli at high repetition rate used for these recordings (20–40 Hz repetition rate for click and tone burst stimulation, respectively). We believe that any sound stimulation and potentially even spontaneous activity may effectively silence

DKO synapses. As expected, longer interstimulus intervals may allow a larger proportion of channels to recover between stimuli, increasing the onset rates of SGN spiking while decreasing the first spike latency and jitter. Still, high-intensity stimulation had to be used in the DKO animals to evoke an increase in SGN spiking. It is possible that spontaneous activity and/or our routines used to determine the SGN characteristic frequency inactivate a majority of IHC Ca$_V$1.3 channels thus not allowing reliable assessment of the characteristic frequency. In addition to Ca$_V$1.3 channel inactivation, other factors may contribute to low hearing sensitivity, such as possible impairment of synaptic vesicle replenishment, or IHC mechanotransduction, and, in some cases, impaired cochlear amplification. A complete lack of BK channels was shown to affect the IHC resting potential, receptor potential amplitude and timing, and consequently the timing and frequency of spiking in the SGNs (*Oliver et al., 2006*). Lower expression of BK channels is not expected to significantly affect hearing thresholds, at least initially (*Rüttiger et al., 2004*; *Maison et al., 2013*; *Pyott et al., 2007*), and thus likely does not contribute to the hearing defect of the *Cabp1/2* DKO animals. However, at advanced ages, the effects described above may combine with the lack of CaBPs to additionally desynchronize already scarce spikes in the auditory nerve fibers.

### Clinical implications for the diagnosis and treatment of DFNB93

The SGN sound encoding phenotype in *Cabp2*-KO mice was surprisingly mild compared to the deficits observed in DFNB93 patients (*Picher, 2017*; *Schrauwen et al., 2012*): SGN thresholds and evoked spike rates were only minimally affected (*Picher, 2017*). Given the strong recovery phenotype in the DKO we now consider the possibility that synaptic activity in response to everyday sounds could already inactivate calcium channels and reduce SGN spiking. In this case, communication in daily life would be more strongly impaired than audiometric data obtained in a soundproof chamber, especially when short stimuli separated by silent intervals are used. In mild cases, a gap detection test may be used to assess auditory fatigue. Based on our data, we further predict that patients that have a stronger channel inactivation phenotype might not benefit much from sound amplification by hearing aids. Instead, they should use technologies that improve the signal to noise ratio (e.g. noise cancellation, directional microphones, and wireless transmitters). An alternative or additional future treatment approach may be gene therapy. In our hands, re-expression of CaBP2 by genetic treatment partially restored Ca$_V$1.3 properties, IHC exocytosis and consequently ABR wave amplitudes and thresholds. This successful viral rescue supports the view that a reversible inactivation of IHC Ca$_V$1.3 currents is the main mechanism of hearing loss in CaBP-related deafness, which can be targeted by gene therapy in the future.

# Materials and methods

**Key resources table**

| Reagent type (species) or resource | Designation | Source or reference | Identifiers | Additional information |
|---|---|---|---|---|
| Strain, strain background (*Mus musculus*) | *Cabp1*-KO | *Kim et al., 2014* | RRID:MGI:5780462 | A KO of *Cabp1* used for cross-breeding to obtain *Cabp1/2*-DKO animals |
| Strain, strain background (*Mus musculus*) | *Cabp2*$^{LacZ/LacZ}$ | *Picher, 2017* | RRID:MGI:6155766 | A KO of *Cabp2* used for cross-breeding to obtain *Cabp1/2*-DKO animals |
| Other | PHP.eB viral capsid | Addgene plasmid | #103005 | Gift of V. Gradinaru |
| Recombinant DNA reagent | Encoding CaBP2-L | NCBI Reference Sequence | NP_038906.2 | Long isoform of mouse CaBP2 |
| Antibody | anti-Myosin VIIa (Mouse monoclonal) | Santa Cruz Biotechnology | Cat#: sc-74516, RRID:AB_2148626 | 1:200 |
| Antibody | anti-K$_{ca}$1.1 (Rabbit polyclonal) | Alomone Labs | Cat#: APC-021, RRID:AB_2313725 | 1:200 |
| Antibody | anti-SK2 (Rabbit polyclonal) | Sigma-Aldrich | Cat#: P0483, RRID:AB_260860 | 1:200 |

*Continued on next page*

*Continued*

| Reagent type (species) or resource | Designation | Source or reference | Identifiers | Additional information |
|---|---|---|---|---|
| Antibody | anti-Ca$_V$1.3 (Rabbit polyclonal) | Alomone Labs | Cat#: ACC-005 RRID:AB_2039775 | 1:100 |
| Antibody | anti-CtBP2 (Mouse monoclonal) | BD Biosciences | Cat#: 612044, RRID:AB_399431 | 1:200 |
| Antibody | anti-Homer1 (Chicken polyclonal) | Synaptic Systems | Cat#: 160006, RRID:AB_2631222 | 1:200 |
| Antibody | anti-Synapsin1/2 (Guinea pig polyclonal) | Synaptic Systems | Cat#: 106004, RRID:AB_1106784 | 1:200 |
| Antibody | anti-RibeyeA (Rabbit polyclonal) | Synaptic Systems | Cat#: 192103, RRID:AB_2086775 | 1:200 |
| Antibody | anti-parvalbumin-α (Guinea pig polyclonal) | Synaptic Systems | Cat#: 195004, RRID:AB_2156476 | 1:200 |
| Antibody | anti-GluA3 (Goat polyclonal) | Santa Cruz Biotechnology | Cat#: sc-7612, RRID:AB_2113895 | 1:200 |
| Antibody | anti-GFP (Chicken polyclonal) | Abcam | Cat#: ab13970, RRID:AB_300798 | 1:200 |
| Antibody | anti-chicken Alexa Fluor 488 (Goat polyclonal) | Thermo Fisher Scientific | Cat#: A-11039, RRID:AB_2534096 | 1:200 |
| Antibody | anti-rabbit Alexa Fluor 488 (Goat polyclonal) | Thermo Fisher Scientific | Cat#: A-11008, RRID:AB_143165 | 1:200 |
| Antibody | anti-guinea pig Alexa Fluor 488 (Goat polyclonal) | Thermo Fisher Scientific | Cat#: A-11073, RRID:AB_2534117 | 1:200 |
| Antibody | anti-rabbit Alexa Fluor 568 (Goat polyclonal) | Thermo Fisher Scientific | Cat#: A-11011, RRID:AB_143157 | 1:200 |
| Antibody | anti-chicken Alexa Fluor 568 (Goat polyclonal) | Abcam | Cat#: ab175711, RRID:AB_2827757 | 1:200 |
| Antibody | anti-mouse Alexa Fluor 647 (Goat polyclonal) | Thermo Fisher Scientific | Cat#: A-21236, RRID:AB_2535805 | 1:200 |
| Antibody | anti-mouse Alexa Fluor 633 (Goat polyclonal) | Thermo Fisher Scientific | Cat#: A-21136, RRID:AB_2535775 | 1:200 |
| Antibody | anti-guinea pig Alexa Fluor 633 (Goat polyclonal) | Thermo Fisher Scientific | Cat#: A-21105, RRID:AB_2535757 | 1:200 |
| Antibody | anti-mouse Alexa Fluor 488 (Donkey polyclonal) | Thermo Fisher Scientific | Cat#: A-21202, RRID:AB_141607 | 1:200 |
| Antibody | anti-goat Alexa Fluor 568 (Donkey polyclonal) | Thermo Fisher Scientific | Cat#: A-11057, RRID:AB_2534104 | 1:200 |
| Antibody | anti-rabbit Alexa Fluor 647 (Donkey polyclonal) | Thermo Fisher Scientific | Cat#: A-31573, RRID:AB_2536183 | 1:200 |
| Software, algorithm | Patchers Power Tools | Igor Pro XOP (http://www3.mpibpc.mpg.de/groups/neher/index.php?page=software) | RRID:SCR_001950SCR_001950 | Analysis of patch-clamp data |
| Software, algorithm | ImageJ | http://imagej.nih.gov/ij/ | RRID:SCR_003070 | Image assembly |
| Software, algorithm | Imaris, Bitplane | http://www.bitplane.com/imaris/imaris | RRID:SCR_007370 | Image assembly and analysis |
| Software, algorithm | Microsoft Excel | https://www.microsoft.com/en-gb/ | RRID:SCR_016137 | Data overviews, algorithm |

*Continued on next page*

*Continued*

| Reagent type (species) or resource | Designation | Source or reference | Identifiers | Additional information |
|---|---|---|---|---|
| Software, algorithm | Fiji | http://fiji.sc | RRID:SCR_002285 | Image assembly |
| Software, algorithm | GraphPad Prism, GraphPad Software | https://graphpad.com | RRID:SCR_002798 | Statistical analyses and graphs |
| Software, algorithm | IgorPro 6, Wavemetrics | http://www.wavemetrics.com/products/igorpro/igorpro.htm | RRID:SCR_000325 | Data analyses, statistical analyses, and graphs |
| Software, algorithm | MATLAB, Mathworks | http://www.mathworks.com/products/matlab/ | RRID:SCR_001622SCR_001622 | Data analyses, statistical analyses, and graphs |
| Software, algorithm | Patchmaster, HEKA, Harvard Bioscience | https://www.heka.com/index.html | - | Data acquisition |
| Software, algorithm | BioSig, TDT | https://www.tdt.com/components/software/ | RRID:SCR_008428 | Data acquisition |
| Software, algorithm | iMSPECTOR, Abberrior Instruments | https://abberior.rocks/superresolution-confocal-systems/software/ | RRID:SCR_015249 | Data acquisition |

## Animals

For the generation of the *Cabp1/Cabp2*-double-knockout (*Cabp1/2*-DKO) line, *CaBP2$^{LacZ/LacZ}$* mice, carrying LacZ trapping cassette in place of the exons 3 and 4 (*Picher, 2017*), were crossbred with *Cabp1* KO mice (*Kim et al., 2014*) and maintained on a C57/Bl6 background. Mice of both sexes from 3 to 13 weeks were used for electrophysiological and systems physiology recordings. All experiments complied with the national animal care guidelines and were approved by the University of Göttingen Board for animal welfare and the animal welfare office of the state of Lower Saxony (AZ 33.19-42502-04-19/3134 and AZ 33.19-42502-04-19/3133).

## Systems physiology recordings: ABR and DPOAE

Prior to the experiments mice were weighed and anesthetized via intraperitoneal injections of ketamine (125 µg/g) and xylazine (2.5 µg/g). Recordings were obtained in a custom-made sound-proof box with a Tucker-Davis (TDT) BioSig System II/BioSig software (Tucker-Davis Technologies, Alachua, FL). Body temperature was maintained using a heating pad (37°C) and ECG was monitored continuously. For ABR recordings, subcutaneous needles were placed at the vertex, the ipsilateral pinna and at the lower back. Using a JBL2402 speaker in free-field configuration, tone bursts (10 ms plateau, 1 ms $cos^2$ rise/fall) or clicks of 0.03 ms were ipsilaterally presented at 40 Hz for tone bursts or 20 Hz for clicks. ABR traces were amplified 50000-fold, band-pass filtered between 400 Hz and 4 kHz (NeuroAmp) and averaged from 1300 repetitions sampled at 50 kHz for 20 ms. Thresholds were defined in stacked waveforms as the lowest stimulus intensity (dB SPL) at which a reproducible waveform could be visually detected with a 10-dB precision. Wave I amplitudes were measured peak-to-trough. DPOAEs were measured using a custom-written MATLAB (MathWorks) routine. Two primary tones f1 and f2 (ratio f1/f2 = 1.2, level difference f2 = f1 + 10 dB) were presented via two speakers (MF-1, TDT) and a custom-made ear probe containing an MKE-2 microphone (Sennheiser). Stimulus duration was 16 ms and f2 levels varied from 10 to 70 dB SPL in 10 dB steps. The microphone signal was amplified (DMX 6Fire; Terratec (young animals), or UAC-2; Zoom (older animals)) and digitized (TDT System III). DPOAE amplitude was analyzed at $2 \times f2 - f1$ using custom-written MATLAB software (MathWorks) with fast Fourier transformation. DPOAE thresholds were determined as responses that exceeded –18 dB SPL in young and –4 dB SPL in older animals. Experimental data with obvious technical abnormalities (higher noise floor, inappropriate selection of frequency/intensity combinations) was excluded.

## Auditory nerve fiber recordings

Mice were anesthetized by injection of xylazine (5 mg/kg), urethane (1.32 mg/kg), and buprenorphine (0.1 mg/kg) i.p., tracheotomized, and placed in a stereotactic apparatus. The left occipital bone and the cerebellum were partially removed to visualize the left superior semicircular canal. The latter

served as a landmark to navigate a glass microelectrode through the cochlear nucleus, aiming at the location where the auditory nerve leaves the internal auditory canal and enters the cochlear nucleus. 85 or 90 dB broadband noise bursts (ScanSpeak, Avisoft Bioacoustics) were used as search stimuli to identify neurons that displayed sound-driven spiking activity. They were characterized by spontaneous rate and frequency tuning and the responses to tone bursts at the characteristic frequency, 30 dB above the threshold, as previously described (*Strenzke et al., 2016*). In the majority of sound-responsive DKO neurons, thresholds were so high and tuning so broad that neither the automated tuning curve algorithm nor systematic or manual changes in sound frequency elicited frequency-specific responses. In these cases, we either chose a subjective 'best frequency' or used white noise burst stimulation at 85 dB.

## Patch-clamp recordings

For patch-clamp recordings, the apical coils of 3- to 4-week-old animals were dissected in HEPES (2-[4-(2-hydroxyethyl)-1-piperazinyl]ethanesulfonic acid)-Hanks solution containing (in mM): 10 HEPES, 5.35 KCl, 141.7 NaCl, 0.5 MgSO$_4$, 1 MgCl$_2$, 11.1 D-glucose and 3.42 L-glutamine. pH was adjusted to 7.2. IHCs of the ~6 kHz tonotopical region were investigated using the perforated patch configuration. The pipette solution contained (in mM): 130 Cs-gluconate, 10 Tetraethylammonium-Cl, 10 4-AP, 1 MgCl$_2$, 10 HEPES, 300 mg/ml amphotericin B, pH 7.2–290 mOsm. The bath solution was composed of (in mM): 111 NaCl, 35 TEA-Cl, 2.8 KCl, 1 CsCl, 1 MgCl$_2$, 10 NaOH–HEPES, 11.3 D-glucose, and 1.3 or 2 CaCl$_2$ or BaCl$_2$ (pH 7.2, 300–310 mOsm). When the bath solution contained 2 mM BaCl$_2$/CaCl$_2$, NaCl was adjusted to 110 mM. The bath solution was supplemented by the SK2 channel blocker apamin (100 nM; Peptanova, Germany) freshly on the day of the experiment. Recordings were performed at room temperature using an EPC-9 amplifier (Heka-Germany) controlled by Pulse software. Currents were leak-corrected using a p/10 protocol, sampled at 50 kHz and corrected offline for the liquid junction potential (–15 mV). IHCs were patched at a holding potential of –85 mV (in a subset of experiments, –65 and –55 mV) and stimulated with depolarizations to the peak Ca$^{2+}$-current potential. Changes in membrane capacitance were monitored by applying a 2-kHz-sine wave with an amplitude of 35 mV. In a subset of experiments, the amplitude was reduced to 15 mV to diminish possible stimulation of the cells caused by the sine wave. Recordings with leak currents exceeding –50 pA, series resistance above 34 MΩ and unstable recordings were excluded from analysis. VDI and CDI were calculated as follows: VDI = 1 − $I_{Ba\_500}/I_{Ba\_initial}$ and CDI = 1 − $(I_{Ca\_500}/I_{Ca\_initial})/(I_{Ba\_500}/I_{Ba\_initial})$, where $I_{Ba\_500}$ and $I_{Ca\_500}$ represent whole-cell Ba$^{2+}$ and Ca$^{2+}$ currents 500 ms after beginning of depolarization, and $I_{Ba\_initial}$ and $I_{Ca\_initial}$ correspond to the initial, peak currents with the respective divalent cations (*Tadross et al., 2010*).

## Immunohistochemistry

Apical coils (6–8 kHz tonotopical region) of the organ of Corti were fixed with 4% (vol/vol) formaldehyde in phosphate-buffered saline (PBS) on ice. After incubation for 1 hr at room temperature in goat serum dilution buffer [16% (vol/vol) normal goat serum, 450 mM NaCl, 0.6% Triton X-100, 20 mM phosphate buffer (pH 7.4)], the explants were incubated in primary antibodies over night at 4°C. The following antibodies were used: chicken anti-Homer1 (1:200, Synaptic Systems), mouse anti-Myosin VIIa (1:200, Santa Cruz), guinea pig anti-Synapsin1/2 (1:200, Synaptic Systems), rabbit anti-RibeyeA (1:200, Synaptic Systems), guinea pig anti-parvalbumin-α (1:200; Synaptic Systems), rabbit anti-BK (1:200; Alomone), rabbit anti-SK2 (1:200; Sigma), mouse anti-CtBP2 (1:200; BD Biosciences), goat anti-GluA3 (1:200; Santa Cruz), rabbit-anti-CaV1.3 (1:100; Alomone), and chicken anti-GFP (1:200, Abcam). The explants were then incubated with secondary AlexaFluor-labeled antibodies (1:200; Molecular Probes/Thermo Fischer, Abcam) for 1 hr at room temperature, washed three times in PBS and mounted. Confocal images were acquired with an Abberior Instruments Expert Line STED microscope using a 1.4 NA ×100 oil immersion objective. ImageJ software (NIH) was employed to analyze 2D confocal stacks and generate maximum projections. Volumetric synapse and BK-cluster immunofluorescence analysis ran on Imaris (Bitplane, Zurich, Switzerland) paired with a custom-written Matlab software (*Ohn et al., 2016*; *Neef et al., 2018*).

## Molecular biology and AAV vector production

The construct containing the long isoform of mouse CaBP2 (CaBP2-L; NCBI Reference Sequence: NP_038906.2) and eGFP is depicted in *Figure 1—figure supplement 1A*. AAV9-PHP.eB particles

were generated using our standard AAV purification procedure previously described in more detail in *Huet and Rankovic, 2021*. In brief, triple transfection of HEK-293T cells was performed using pHelper plasmid (TaKaRa/Clontech), the *trans*-plasmid providing viral capsid PHP.eB (a generous gift from Viviana Gradinaru, Addgene plasmid #103005) and the *cis*-plasmid providing hCMV/HBA_wtCaBP2-P2A-eGFP. PHP.eB viral particles were harvested from the medium 72 hr after, and from cells and the medium 120 hr after transfection. Precipitation was done with 40% polyethylene glycol 8000 (Acros Organics, Germany) in 500 mM NaCl for 2 hr at 4°C. Both, precipitate and cells were lysed in high salt buffer (500 mM NaCl, 2 mM $MgCl_2$, 40 mM Tris–HCl, pH 8.0) and non-viral DNA was degraded using salt-activated nuclease (SAN, Arcticzymes, USA). The cell lysates were clarified by centrifugation at 2000 × $g$ for 10 min and then purified over iodixanol (Optiprep, Axis Shield, Norway) step gradients (15, 25, 40, and 60%) at 350,000 × $g$ for 2.25 hr (*Grieger et al., 2006*; *Zolotukhin et al., 1999*). Finally, viral particles were concentrated using Amicon filters (EMD, UFC910024) and formulated in sterile PBS supplemented with 0.001% Pluronic F-68 (Gibco, Germany). Virus titer was 8.2 × $10^{12}$ genome copies/ml measured by determining the number of DNase I resistant genome copies using qPCR (StepOne, Applied Biosystems) and AAV titration kit (TaKaRa/Clontech). The purity of viruses was routinely checked by silver staining (Pierce, Germany) after gel electrophoresis (NovexTM 4–12% Tris-Glycine, Thermo Fisher Scientific).

### Virus injections

Mice were injected at postnatal day 6 using the round window approach as described in earlier studies (*Rankovic et al., 2020*). In brief, anesthesia was established with isoflurane (5% for induction, 2–3% for maintenance, frequent testing of the absence of hind-limb withdrawal reflex). For analgesia, buprenorphine (0.1 mg/kg, 30 min before surgery) and carprofen (5 mg/kg, during and 1 day after surgery) were applied subcutaneously and Xylocain (10 mg spray) locally. Body temperature was maintained by a custom-built heating blanket. Following a retro-auricular approach, the facial nerve was exposed to determine where to puncture the cartilaginous bulla with the injection pipette and target the scala tympani where virus suspension (1–1.5 µl, corresponding to 0.82–1.23 × $10^{10}$ AAV particles) was injected. Following injection, the endogenous tissue was relocated and the surgical situs closed by suturing the skin. Two to three weeks later, the injected mice were used for hearing tests, in vitro electrophysiology and immunohistochemistry of the cochleae.

### Data analysis

Data were analyzed using Igor Pro Software (Wavemetrics), GraphPad PRISM (GraphPad Software, Inc), Matlab (Mathworks), and Imaris (Bitplane). Mean membrane capacitance changes and calcium current estimates present grand averages calculated from the mean estimates of individual IHCs. All data are presented as mean ± SEM. For the analysis of half-max $Ca^{2+}$ current ($V_{half}$) and activation constant ($k_{act}$), current–voltage traces were transformed into an activation function ($I_{Ca} = g_{max}(V − V_{rev})$), which was subsequently fitted by a Boltzmann function: $1/(1 + \exp((V_{0.5} − V)/k_{act}))$. $G_{max}$ describes the maximum conductance, $V$ the holding potential and $V_{rev}$ the reversal potential. Data were tested for randomness, normality (Jaque–Bera test), equality of variances ($F$-test) and finally compared for statistical significance by performing Student's $t$-test for normally distributed data of equal variance or else by the Wilcoxon rank-sum test. Systems physiology data were tested for significance using ANOVA with Tukey's or Šidak's multicomparisons tests. For each dataset, sample sizes were decided according to typical sizes in the respective fields (e.g. systems or cell electrophysiology, and immuno-histochemistry of ribbon synapses). Respective sections, figures, or figure legends report the number of biological replicates ($n$), animal numbers ($N$), and the statistical tests used.

### Materials availability statement

Research materials and biological reagents used in this MS are reported in the Materials and Methods section. The custom routines and scripts used in the MS are provided as Source Codes or within data source files. Custom routines for analysis of patch-clamp data (IVs, inactivation, exocytosis, recovery from inactivation) are available within source files: *Figure 1—source data 1*, *Figure 1—source data 2*, *Figure 3—source data 2*, and *Figure 4—source data 1*. Other source codes: MATLAB scripts for DPOAE analysis (related to *Figure 6—figure supplement 1*), single unit recordings (related to *Figure 6*), and GFP expression analysis (related to *Figure 6—figure supplement 2*). Further source

codes are provided: *Figure 6—figure supplement 1—source code 1* (MATLAB scripts for analysis of DPOAE traces), *Figure 6—figure supplement 2—source code 1* (MATLAB scripts for analysis of GFP immunofluorescent signals) and (MATLAB scripts for analysis of SU data).

## Acknowledgements

The authors would like to thank Christiane Senger-Freitag for the virus injections, Daniela Gerke for help with virus production, and Sandra Gerke, Ina Preuss, and Sina Langer for excellent technical support. The authors would further like to thank Amy Lee (University of Iowa) for providing the *Cabp1* KO animals, and Mohona Mukhopadhyay and Tobias Moser for critical reading of the paper, useful comments and suggestions. This work was supported by the German Research Foundation: DFG Priority Program 1608 'Ultrafast and temporally precise information processing: Normal and dysfunctional hearing' [to TP (PA 2769/1-1)], the Clinician Scientist Program 'Cell Dynamics in Disease and Therapy' at the University Medical Center Göttingen (to DO; project number 413501650), CRC889 (subproject A06; to NS, and B09; to TP); and by the Multiscale Bioimaging Cluster of Excellence (EXC 2067/1-390729940; to TP).

## Additional information

### Funding

| Funder | Grant reference number | Author |
| --- | --- | --- |
| Deutsche Forschungsgemeinschaft | PP1608: PA 2769/1-1 | Tina Pangrsic |
| Deutsche Forschungsgemeinschaft | 413501650 | David Oestreicher |
| Deutsche Forschungsgemeinschaft | CRC889: B09 | Tina Pangrsic |
| Deutsche Forschungsgemeinschaft | CRC889: A06 | Nicola Strenzke |
| Multiscale Bioimaging Cluster of Excellence | EXC 2067/1-390729940 | Tina Pangrsic |

The funders had no role in study design, data collection, and interpretation, or the decision to submit the work for publication.

### Author contributions

David Oestreicher, Conceptualization, Data curation, Formal analysis, Funding acquisition, Validation, Investigation, Visualization, Writing – original draft, Writing – review and editing, Execution of in vitro cell electrophysiology, immunohistochemistry, and ABR/DPOAE recordings from young and injected animals; Shashank Chepurwar, Software, Formal analysis, Investigation, Writing – review and editing, Execution of in vivo single unit recordings and the ABR/DPOAE recordings in older animals under the supervision of NS; Kathrin Kusch, Resources, Methodology, Writing – review and editing, Generation of the viral vectors; Vladan Rankovic, Resources, Methodology, Writing – review and editing, Generation of the viral vectors; Sangyong Jung, Investigation, Writing – review and editing, Execution of the initial injection experiments; Nicola Strenzke, Data curation, Software, Formal analysis, Supervision, Funding acquisition, Validation, Visualization, Writing – original draft, Writing – review and editing; Tina Pangrsic, Conceptualization, Data curation, Software, Formal analysis, Supervision, Funding acquisition, Validation, Visualization, Writing – original draft, Project administration, Writing – review and editing

### Author ORCIDs

David Oestreicher ⓘ http://orcid.org/0000-0002-4541-6398
Nicola Strenzke ⓘ https://orcid.org/0000-0003-1673-1046
Tina Pangrsic ⓘ https://orcid.org/0000-0002-7313-1648

### Ethics

All experiments complied with the national animal care guidelines and were approved by the University of Göttingen Board for animal welfare and the animal welfare office of the state of Lower Saxony (AZ 33.19-42502-04-19/3134 and AZ 33.19-42502-04-19/3133).

Reviewer #1 (Public Review): https://doi.org/10.7554/eLife.93646.3.sa1
Reviewer #2 (Public Review): https://doi.org/10.7554/eLife.93646.3.sa2
Reviewer #3 (Public Review): https://doi.org/10.7554/eLife.93646.3.sa3
Author response https://doi.org/10.7554/eLife.93646.3.sa4

## Additional files

### Supplementary files

• Supplementary file 1. Statistical analysis of ABR thresholds and ABR wave I–V latencies. The table lists the adjusted p-values of the Tukey's multicomparisons tests upon one-way (20 Hz clicks) or two-way ANOVA (otherwise) for the ABR thresholds and the click-evoked ABR wave I–V latencies. Very low p-values are represented by asterisks: $p < 0.001$ (***), $p < 0.0001$ (****).

• MDAR checklist

### Data availability

All data generated or analyzed during this study are included in the manuscript and supporting files; source data files have been provided for Figures 1–7, and figure supplements.

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
