## [Editor Report · eLife assessment]

This **fundamental** work substantially advances our understanding of the role of calcium-binding proteins 1 and 2 (CaBP1 and CaBP2) for generating sustained calcium currents in mouse inner hair cells and their capacity for indefatigable exocytosis. The evidence supporting the conclusions is **compelling**, with rigorous in vitro and in vivo physiological experiments and state-of-the-art microscopy. The work will be of broad interest to synaptic physiologists, cellular biochemists, and hearing researchers.

---

## [Referee Report · Reviewer #1 (Public Review)]

Summary:

This manuscript dissects the contribution of the CaBP 1 and 2 on the calcium current in the cochlear inner hair cells. The authors measured the calcium current inactivation from the double knock-out CaBP1 and 2 and show that both proteins contribute to the voltage-dependent and calcium-dependent inactivation. Synaptic release was reduced in the double KO. As a consequence, the authors observed a depressed activity within the auditory nerve. Taken together, this study identifies a new player that regulates the stimulation-secretion coupling in the auditory sensory cells.

Strengths:

In this study, the authors bring compelling evidence that CaBP 1 and 2 are both involved in the inactivation of the calcium current, from cellular up to system level and by taking care to probe different experimental conditions such as different holding potentials and by rescuing the phenotype with the re-expression of CaBP2. Indeed, while changing the holding potential worsen the secretion, it completely changes the kinetics of the inactivation recovery. It alerts the reader that probing different experimental conditions that may be closer to physiology are better suited to uncover any deleterious phenotype. This gave pretty solid results.

Weaknesses:

Although this study clearly points that CaBP1 is involved in the calcium current inactivation, it is not clear how CaBP1 and CaBP2 act together (but this is probably beyond the scope of the study). Another point is that the authors re-express CaBP2 to largely rescue the phenotype in the double KO but no data are available to know whether the re-expression of both CaBP1 and CaBP2 would achieve a full recovery and what would be the effect of the sole re-expression of CaBP1 in the double KO.

---

## [Referee Report · Reviewer #2 (Public Review)]

Summary:

In the manuscript by Oestreicher et al, the authors use patch-clamp electrophysiology, immunofluorescent imaging of the cochlea, auditory function tests, and single-unit recordings of auditory afferent neurons to probe the unique properties of calcium signaling in cochlear hair cells that allow rapid and sustained neurotransmitter release. The calcium binding proteins (CaBPs) are thought to modify inactivation of the Cav1.3 calcium channels in IHCs that initiate vesicle fusion, reducing the calcium-dependent inactivation (CDI) of the channels to allow sustained calcium influx to support neurotransmitter release. The authors use knockout mice of Cabp1 and Cabp2 in a double knockout (Cabp1/2 DKO) to show that these molecules are required for enabling sustained calcium currents by reducing CDI, enabling proper IHC neurotransmitter release. They further support their evidence by re-introducing Cabp2 using injection of AAV containing the Cabp2 sequence into the cochlea, which restores some of the auditory function and reduces CDI in patch-clamp recordings.

Strengths:

Overall the data is convincing that Cabp1/2 is required for reducing CDI in cochlear hair cells, allowing their sustained neurotransmitter release and sound encoding. Figures are well-prepared, recordings are careful and stats are appropriate, and the manuscript is well written. The discussion appropriately considers aspects of the data that are not yet explained and await further experimentation.

Weaknesses:

There are some sections of the manuscript that pool data from different experiments with slightly different conditions (wt data from a previous paper, different calcium concentrations, different holding voltages, tones vs clicks, etc). This makes the work harder to follow and more complicated to explain. However, the major conclusion, that that cabp1 and 2 work together to reduce calcium dependent inactivation of L-type calcium channels in cochlear inner hair cells, still holds and is well supported. Another minor weakness is that the authors used injections of AAV containing sequences for Cabp2, but do not present data from sham surgeries. In most cases, the improvement of hearing function with AAV injection is believable and should be attributed to the cabp2 function. However, in at least one instance (Figure 4B), the results of the AAV injection experiments may be overinterpreted - the authors show that upon AAV injection, the hair cells have a much longer calcium current recovery following a large, long depolarization to inactivate the calcium channels. Without comparison to a sham surgery, it is not known if this result could be a subtle result of the surgery or indeed due to the Cabp2 expression. The authors have added text acknowledging this, as appropriate.

---

## [Referee Report · Reviewer #3 (Public Review)]

Summary:

The authors attempted to unravel the role of the Ca2+-binding proteins CaBP1 and CaBP2 for the hitherto enigmatic lack of Ca2+-dependent inactivation of Ca2+ currents in sensory inner hair cells (IHCs). As Ca2+ currents through Cav1.3 channels are crucial for exocytosis, the lack of inactivation of those Ca2+ currents is essential for the indefatigable sound encoding by IHCs. Using a deaf mouse model lacking both CaBP1 and CaBP2, the authors convincingly demonstrate that both CaBP1 and CaBP2 together confer a lack of inactivation, with CaBP2 being far more effective. This is surprising given the mild phenotype of the single knockouts, which has been published by the authors before. Re-admission of CaBP2 through viral gene transfer into the inner ear of double-knockout mice largely restored hearing function, normal Ca2+ current properties, and exocytosis.

Comments on the revised version:

The authors improved the quality of the figures as requested.

---

## [Author Response]

The following is the authors’ response to the original reviews.

**Public Reviews:**

**Reviewer #1 (Public Review):**
Summary:This manuscript dissects the contribution of the CaBP 1 and 2 on the calcium current in the cochlear inner hair cells. The authors measured the calcium current inactivation from the double knock-out CaBP1 and 2 and showed that both proteins contribute to voltage-dependent and calcium-dependent inactivation. Synaptic release was reduced in the double KO. As a consequence, the authors observed a depressed activity within the auditory nerve. Taken together, this study identifies a new player that regulates the stimulation-secretion coupling in the auditory sensory cells.Strengths:In this study, the authors bring compelling evidence that CaBP 1 and 2 are both involved in the inactivation of the calcium current, from cellular up to system level, and by taking care to probe different experimental conditions such as different holding potentials and by rescuing the phenotype with the re-expression of CaBP2. Indeed, while changing the holding potential worsens the secretion, it completely changes the kinetics of the inactivation recovery. It alerts the reader that probing different experimental conditions that may be closer to physiology is better suited to uncovering any deleterious phenotype. This gave pretty solid results.Weaknesses:Although this study clearly points out that CaBP1 is involved in the calcium current inactivation, it is not clear how CaBP1 and CaBP2 act together (but this is probably beyond the scope of the study). Another point is that the authors re-express CaBP2 to largely rescue the phenotype in the double KO but no data are available to know whether the re-expression of both CaBP1 and CaBP2 would achieve a full recovery and what would be the effect of the sole re-expression of CaBP1 in the double KO.

We would like to thank the reviewer for the appreciation of our work. We agree that the effect of the sole re-expression of CaBP1 in the double KO remains elusive and have planned to address this question in a follow-up study.

**Reviewer #2 (Public Review):**
Summary:In the manuscript by Oestreicher et al, the authors use patch-clamp electrophysiology, immunofluorescent imaging of the cochlea, auditory function tests, and single-unit recordings of auditory afferent neurons to probe the unique properties of calcium signaling in cochlear hair cells that allow rapid and sustained neurotransmitter release. The calcium-binding proteins (CaBPs) are thought to modify the inactivation of the Cav1.3 calcium channels in IHCs that initiate vesicle fusion, reducing the calcium-dependent inactivation (CDI) of the channels to allow sustained calcium influx to support neurotransmitter release. The authors use knockout mice of Cabp1 and Cabp2 in a double knockout (Cabp1/2 DKO) to show that these molecules are required for enabling sustained calcium currents by reducing CDI and enabling proper IHC neurotransmitter release. They further support their evidence by re-introducing Cabp2 using an injection of AAV containing the Cabp2 sequence into the cochlea, which restores some of the auditory function and reduces CDI in patch-clamp recordings.Strengths:Overall the data is convincing that Cabp1/2 is required for reducing CDI in cochlear hair cells, allowing their sustained neurotransmitter release and sound encoding. Figures are well-prepared, recordings are careful and stats are appropriate, and the manuscript is well-written. The discussion appropriately considers aspects of the data that are not yet explained and await further experimentation.Weaknesses:There are some sections of the manuscript that pool data from different experiments with slightly different conditions (wt data from a previous paper, different calcium concentrations, different holding voltages, tones vs clicks, etc). This makes the work harder to follow and more complicated to explain. However, the major conclusion, that cabp1 and 2 work together to reduce calcium-dependent inactivation of L-type calcium channels in cochlear inner hair cells, still holds.Another weakness is that the authors used injections of AAV-containing sequences for Cabp2, but do not present data from sham surgeries. In most cases, the improvement of hearing function with AAV injection is believable and should be attributed to the cabp2 function. However, in at least one instance (Figure 4B), the results of the AAV injection experiments may be overinterpreted - the authors show that upon AAV injection, the hair cells have a much longer calcium current recovery following a large, long depolarization to inactivate the calcium channels. Without comparison to sham surgery, it is not known if this result could be a subtle result of the surgery or indeed due to the Cabp2 expression. It would be great to see the auditory nerve recordings in AAV-injected animals that have a recovery of ABRs. However, this is a challenging experiment that requires considerable time and resources, so is not required.

We would like to thank the reviewer for the appreciation of our work. We agree with the reviewer that sham surgery may convey more information that might benefit the interpretation of our data. The recovery experiments were very tedious and these long patch-clamp paradigms required extremely stable recordings. Based on our observations, we plan to address the recovery kinetics into more detail in the follow-up study. However, we would consider off-side effects of the surgery (as it may mainly affect middle ear function) and of the empty AAV-vector on inner hair cell calcium current recovery rather unlikely, but we cannot exclude them. We thus added a sentence in the discussion to alert to that. Based on previously published data of the effect of PHP.eB-Cabp2eGFP in WT animals we expect some (mild) adverse effects on hearing from overexpression of CaBP2 and/or eGFP in the inner ear. In the future, we thus plan to further optimize the treatment. In terms of the in vivo recordings from the auditory nerve fibers of the rescued mice, we could not agree more. That is in plan for the follow-up study.

**Reviewer #3 (Public Review):**
Summary:The authors attempted to unravel the role of the Ca2+-binding proteins CaBP1 and CaBP2 for the hitherto enigmatic lack of Ca2+-dependent inactivation of Ca2+ currents in sensory inner hair cells (IHCs). As Ca2+ currents through Cav1.3 channels are crucial for exocytosis, the lack of inactivation of those Ca2+ currents is essential for the indefatigable sound encoding by IHCs. Using a deaf mouse model lacking both CaBP1 and CaBP2, the authors convincingly demonstrate that both CaBP1 and CaBP2 together confer a lack of inactivation, with CaBP2 being far more effective. This is surprising given the mild phenotype of the single knockouts, which has been published by the authors before. Readmission of CaBP2 through viral gene transfer into the inner ear of double-knockout mice largely restored hearing function, normal Ca2+ current properties, and exocytosis.Strengths:(1) In vitro electrophysiology: perforated patch-clamp recordings of Ca2+/Ba2+ currents of inner hair cells (IHCs) from 3-4 week-old mice - very difficult recordings - necessary to not interfere with intracellular Ca2+ buffers, including CaBP1 and CaBP2.(2) Capacitance (exocytosis) recordings from IHCs in perforated patch mode.(3) The insight that a negative holding potential might underestimate the impact of lack of CaBP1/2 on the inactivation of ICa in IHCs. As the physiological holding potential is much more positive than a preferred holding potential in patch clamp experiments it has a strong impact on inactivation in the pauses between depolarization mimicking receptor potentials. This truly advances our thinking about the stimulation of IHCs and accumulating inactivation of the Cav1.3 channels.(4) Insight that the voltage sine method with usual voltage excursions (35 mV) to determine the membrane capacitance (for exocytosis measurements) also favors the inactivated state of Cav1.3 channels(5) Use of double ko mice (for both CaBP1 and CaBP2, DKO) and use of DKO with virally injected CaBP2eGFP into the inner ear.(6) Use of DKO animals/IHCs/SGNs after virus-mediated CaBP2 gene transfer shows a great amount of rescue of the normal ICa inactivation phenotype.(7) In vivo measurements of SGN AP responses to sound, which is highly demanding.(8) In vivo measurements of hearing thresholds, DPOAE characteristics, and ABR wave I amplitudes/latencies of DKO mice and DKO+injected mice compared to WT mice.Very thorough analysis and presentation of the data, excellent statistical analysis.The authors achieved their aims. Their results fully support their conclusions. The methods used by the authors are state-of-the-art.The impacts on the field are the following:Regulation of inactivation of Cav1.3 currents is crucial for the persistent functioning of Cav1.3 channels in sensory transduction.The findings of the authors better explain the phenotype of the human autosomal recessive DFNB93, which is based on the malfunction of CaBP2.Future work - by the authors or others - should address the molecular mechanisms of the interaction of CaBP1 and 2 in regulating Cav1.3 inactivation.Weaknesses:I do not see weaknesses.What is not explained (but was not the aim of the authors) is how the CaBPs 1 and 2 interact with the Cav1.3 channels and with each other to reduce CDI. Also, why DFNB93, which is based on mutation of the CaBP2 gene, lead to a severe phenotype in humans in contrast to the phenotype of the CaBP2 ko mouse.

We would like to thank the reviewer for the appreciation of our work and the amount of effort that went into these experiments. These are the questions that we are posing ourselves as well and would like to address them in the future.

**Recommendations for the authors:**

**Reviewing editor:**
In the Introduction, the authors may also mention that Ca2+-dependent and voltage-dependent inactivation of L-type Ca channels has been reported at ribbon synapses of retinal bipolar cells (see von Gersdorff & Mathtews, J Neurosci. 1996, 16(1):115-122). These are critical retinal interneurons involved in the continuous exocytosis of synaptic vesicles onto retinal ganglion cells.

We would like to thank the reviewing editor for pointing that out, we have added the reference in the revised version of the manuscript.

**Reviewer #1 (Recommendations For The Authors):**
Conditions worsen with age but no numbers regarding the threshold shift are provided.

For better readability, we now included click threshold values for both genotypes and age groups in the MS text, results section.

Do the authors correlate the re-expression level of CaBP2 using GFP to the rescuing phenotype (for exocytosis or BK channels immunostaining)?

The restoration of BK expression in the virus-treated IHC was a side observation of our study, which was not performed in sufficient replicates for proper quantification. In the future, we will address this question into greater detail, possibly with improved viral constructs. In a previous study, we attempted to correlate eGFP fluorescence intensity with residual depolarization-evoked calcium current in CaBP2-injected IHC of Cabp2 single KO animals. At that time, we were unable to establish a convincing correlation. This could be related to (i) large variability in the data, possibly requiring much larger datasets to observe potential correlation above the noise, (ii) variable imaging conditions from prep to prep, or (iii) additional parameters that could influence the outcome of the current rescue, e.g. uncontrolled expression of the transgene. However, we did analyse the correlation between ABR click thresholds and mean IHC eGFP fluorescence in another, preliminary set of data that included different viruses at different titres. There, we were able to observe a relatively good correlation. Interestingly, some of the highest expression levels resulted in poorer threshold recovery, which could indicate harmful overexpression. Moreover, the correlation was only detected when the difference of the mean eGFP expression levels per organ was large. Furthermore, significantly less efficient ABR threshold recovery was observed in the non-injected contralateral ears, which showed a significantly lower viral expression of the transgene. In our follow-up study, we will investigate the question of dose dependence of rescue in more detail.

**Reviewer #2 (Recommendations For The Authors):**
- There are two paragraphs in the results text about supplemental figure #2, which suggests that it should be moved to the main figures.

We would like to thank the reviewer for this suggestion. Figure S2 has now been moved to the main figures (as current Figure 5) and has been modified to accommodate the BK cluster analysis panel. The histogram with the number of ribbon synapses was removed as the data was redundant with the numbers given in the MS text.

- Overall it is hard to distinguish between dark blue and black in many figures, including the dual-color asterisks.

To improve the readability and clarity of the figures, we exchanged dark blue with magenta. Dual-color asterisks in Fig. 3 were changed to single-color asterisks and what they refer to is explain in the figure legend.

- Figure 4 legend - there is a mis-spelling of cabp in the fourth line from the bottom.- Figure 4 legend - the last line does not make sense - describes recovery as being both 'much faster' and 'slowest'.- Figure 6 title - consider removing 'nearly blocked' and replacing it with 'impaired'.

We would like to thank the reviewer for noticing these mistakes that have been corrected in the revised version, as suggested.

- The calculations of VDI and CDI could be better explained, specifically detailing that VDI is calculated first from currents using barium as a divalent, followed by the calculation of CDI.

We included an explanatory sentence in the results section as suggested and are additionally referring the readers to the methods section for the mathematical formulas.

- Why were two different tests (one parametric and one non-parametric) used for the Figure 3B data?

We performed a point-by-point-comparison of data. The choice of test was made based on the distribution and the variance of the data points. We now opted for a unified test, t test with Welch correction, which assumes that samples come from populations with normal distribution, but does not make assumption about equal variances. The outcome of these tests were similar.

- The much broader tuning of the auditory nerve fibers is interesting, consider including this in a figure.

For recording tuning curves, we use an automated algorithm which adapts the tone burst intensity and frequency depending on the preceding results. The threshold criterion is an increase of spiking by 20Hz above spontaneous rate. This routine works fairly well in wild-type animals. However, DKO SGNs typically had very high thresholds at >80 dB across all frequencies, which can partly be explained by the fact that they had very low spike rates and did not reach that criterion. Besides tuning curve runs, we also tried systematic frequency sweeps and manual frequency control to determine a best frequency, followed by a rate intensity function at that frequency to determine “best threshold”.

All this was difficult, because in the DKO SGNs, sound threshold detection was challenged by the strong dependence of spiking on the duration of the preceding silent interval. A preceding stimulus outside the frequency response area or below the activation threshold of the SGN would thus improve spiking by allowing for longer recovery, while a preceding efficient stimulus would reduce it. Thus, the sound threshold determined in a rate level sweep varied depending on the interstimulus interval and possibly even on the (randomized) order at which the intensities were played.

A meaningful threshold measure would require long silent interstimulus intervals, i.e. a long recording time. As tuning curves require multiple threshold measures, it seemed impossible to obtain a useful dataset at high quality. As we deemed the spike rate dependence on interstimulus intervals more important than the tuning we rather focused on tone burst responses acquired at frequency/intensity combinations at which the hair cells and their synapses were maximally activated. In wild-types, these would be tone bursts at characteristic frequency or noise bursts in the saturated part of the rate intensity function, which typically has a dynamic range of 10-25dB. As we assume (based on DPOAE) that cochlear micromechanics and amplification are mostly normal in the DKOs, we hypothesize that the sensitivity and dynamic range of basilar membrane motion and inner hair cell transduction are normal and that the increase in single unit thresholds and loss of sharp tuning are another readout of synaptic dysfunction.

- Figure S2 - please show separate panels for each channel, it is very difficult to make out the changes by eye in the merged panels.

Done.

- Figure S2 G - the results text stated that the BK channel clusters 'appeared' smaller - why was this not measured?

We have performed additional experiments to enable proper analysis of the BK channel clusters. The analysed data shows that the BK clusters are considerably larger and more abundant in the WT as compared to CaBP1/2-deficient IHCs of approx. 4-week-old mice. The results of the analysis are included in the immunohistochemistry figure (now Fig. 5) and are further commented in the results section.

**Reviewer #3 (Recommendations For The Authors):**
I have only a few minor points on the MS:(1) Some labels in Figure 1 are too small and hard to read, e.g. y-axis in B-F. Wherever you use subscripts on the axes, the labeling needs to be larger.(2) Fig. 1A: the colors for CaM and CaBP1.2 are too similar, at least on my printout. Please use more distant colors.(3) Reference 24 should be corrected (no longer in press).

These points have been addressed in the revised version of the MS.